# Microbes translocation from oral cavity to nasopharyngeal carcinoma in patients

Ying Liao [1,4], Yan-Xia Wu[1,4], Minzhong Tang[2], Yi-Wei Chen[1], Jin-Ru Xie[1], Yan Du[1], Tong-Min Wang [1], Yong-Qiao He[1], Wen-Qiong Xue[1], Xiao-Hui Zheng[1], Qiao-Yun Liu[3], Mei-Qi Zheng[1], Yi-Jing Jia[3], Xia-Ting Tong[3], Ting Zhou [1], Xi-Zhao Li[1], Da-Wei Yang[3], Hua Diao[3] & Wei-Hua Jia [1,3] ✉

The presence of oral microbes in extra-oral sites is linked to gastrointestinal cancers. However, their potential ectopically colonization in the nasopharynx and impact on local cancer development remains uncertain. Our study involving paired nasopharyngeal-oral microbial samples from nasopharyngeal carcinoma (NPC) patients and controls unveils an aberrant oral-to-nasopharyngeal microbial translocation associated with increased NPC risk (OR = 4.51, $P$ = 0.012). Thirteen species are classified as oral-translocated and enriched in NPC patients. Among these, *Fusobacterium nucleatum* and *Prevotella intermedia* are validated through culturomics and clonal strain identification. Nasopharyngeal biopsy meta-transcriptomes confirm these microbes within tumors, influencing local microenvironment and cytokine response. These microbes correlate significantly with the Epstein-Barr virus (EBV) loads in the nasopharynx, exhibiting an increased dose-response relationship. Collectively, our study identifies oral microbes migrating to the nasopharynx, infiltrating tumors, impacting microenvironments and linking with EBV infection. These results enhance our understanding of abnormal microbial communication and their roles in carcinogenesis.

Trillions of microbes inhabit both inside and outside of the body, including the oral cavity, gastrointestinal tract, respiratory tract, skin, etc. Specific parts of our body possess specialized microbial communities deemed to be essential for maintaining human health[1,2]. However, heterologous microbes could colonize other adjacent or distant tissues ectopically, altering the local microenvironment and increasing the risk of diseases, particularly the translocations of the oral microbiota[3,4]. For example, some specific oral microbes, such as *Fusobacterium nucleatum*, could overcome physical and/or chemical barriers to ectopically accumulate in the gut under pathological conditions, and participate in the pathogenesis of inflammatory bowel disease and colorectal cancer[5–7].

Similarly, the interactions between oral and pulmonary microbiota have been recently identified, with the enrichment of oral microbes in the lung related to pulmonary pro-inflammatory phenotype and worse prognosis in lung cancer patients[8,9]. The nasopharynx is the gateway of the upper airway and an important niche for microbiota. In general, the oral cavity and nasopharynx maintain different ecological niches with various commensal residents. The loss of niche differences between these two sites has been observed in some upper respiratory tract infections[10]. The interactions between oral and nasopharyngeal microbiota and their association with other disease phenotypes, such as cancers, remain to be elucidated.

[1]State Key Laboratory of Oncology in South China, Collaborative Innovation Center for Cancer Medicine, Guangdong Key Laboratory of Nasopharyngeal Carcinoma Diagnosis and Therapy, Sun Yat-sen University Cancer Center, Guangzhou, China. [2]Key Laboratory of Nasopharyngeal Carcinoma Molecular Epidemiology, Wuzhou Red Cross Hospital, Wuzhou, Guangxi, China. [3]School of Public Health, Sun Yat-sen University, Guangzhou, China. [4]These authors contributed equally: Ying Liao, Yan-Xia Wu. ✉e-mail: jiawh@sysucc.org.cn

Nasopharyngeal carcinoma (NPC) is a malignant tumor arising from the nasopharyngeal mucosa[11]. According to the International Agency for Research on Cancer, in 2020, there were about 133 thousand new cases of NPC, accounting for 0.7% of all cancer diagnoses[12]. Epstein-Barr virus (EBV) infection, host genetics and environmental factors are closely associated with the occurrence and progression of NPC. Recently, the clinical significance of nasopharyngeal microbiota in NPC is starting to gain attention since the local microbiota seem to be associated with NPC therapy response, side effects and prognosis[13–15]. However, the detailed characterization of nasopharyngeal microbiota, its' association with oral microbiota, the tripartite interactions among nasopharyngeal microbiota, host and EBV, and its role in carcinogenesis remain unclear. Particularly, what alterations occurred in nasopharyngeal and oral microbiota between normal people and NPC patients? How do these microbiotas in different locations communicate? Could the nasopharyngeal microbiota interact with EBV infection to promote the onset and progression of NPC?

To answer these questions, we included 148 NPC patients and 124 non-malignant controls from two independent cohorts, comprising 272 pairs of nasopharyngeal-oral samples. Using 16S rRNA gene sequencing, targeted V1-V9 regions (PacBio) and V4 region (Illumina PE250), we profiled the nasopharyngeal and oral microbiota alterations and characterized their communication pattern specific to NPC patients. The microbial culturomics was performed to verify the oral-to-nasopharyngeal translocated taxa specific to NPC. The meta-transcriptomic data from 101 nasopharyngeal tissues demonstrated the intratumor localization and the involvement of these translocated oral microbes in the tumor microenvironment remodeling in NPC patients. Their correlations with EBV loads in nasopharyngeal mucosa further suggested the possible roles of these translocated microbes in the pathogenesis of NPC via facilitating epithelial EBV infection. Taken together, our data uncovered the abnormal translocated microbes from the oral cavity to reshape the nasopharyngeal microecology in NPC patients, which might participate in NPC pathogenesis.

## Results

### Characteristics of study cohorts and sequencing data

To explore the profiles of nasopharyngeal and oral microbiota associated with NPC, we recruited two independent cohorts from two provinces with the highest incidence of NPC in Southern China, and collected their paired nasopharyngeal-oral samples (Fig. 1a). For Cohort 1, the full-length of 16S rRNA gene was sequenced using PacBio long-read SMRT technology. Totally, 6,036,007 high-quality reads with an expected error rate of 0.089% per base were obtained. After denoising, 70 patients and 86 controls with sufficient amplicon sequence variants (ASVs) were retained for subsequent study (Supplementary Fig. 1a–c and Supplementary Table 1). Overall, a median of 91.02% and 80.73% of the ASVs were accurately assigned to species in nasopharyngeal and oral microbial data, providing high-fidelity species-level information for analysis (Supplementary Fig. 1d). As for Cohort 2, the microbial data were obtained through Illumina 16S rRNA gene V4-region sequencing. 78 patients and 38 controls with eligible data were finally included (Supplementary Fig. 1a–c and Supplementary Table 1). As expected, the V4-region data was not sufficient for species-level analysis (Supplementary Fig. 1d). Therefore, we used genus-level data, which had a median assignation rate of 97.46%, for the subsequent analysis instead.

### The translocation of oral microbes to nasopharynx shaped the characteristic microbial profiles associated with NPC

We first assessed the differences between nasopharyngeal and oral microbial communities. Nasopharyngeal microbiota showed lower diversity when compared with oral microbiota (Fig. 1b and Supplementary Fig. 2a, b). The microbial compositions of these two sites were significantly different, which were clearly separated in the principal

coordinate analysis (PCoA) plot (Supplementary Fig. 2c, d). The dominant genera were distinct between the two sites and the relative abundance of the top 10 taxa were clearly different, for example, *Corynebacterium*, *Staphylococcus* and *Cutibacterium* in the nasopharynx, while *Streptococcus*, *Neisseria* and *Prevotella* in the oral cavity (Supplementary Fig. 3). These revealed that the nasopharynx and oral cavity owned distinct microbial niches. Interestingly, we observed that points from two sites were closer in NPC patients (upper plate) than controls (lower plate) in both cohorts in the PCoA plots (Fig. 1c and Supplementary Fig. 4a). Meanwhile, the compositional similarities between the two sites were significantly higher in NPC patients than in controls (Fig. 1d and Supplementary Fig. 4b). We further observed the abundances of potential oral pathogens, including taxa belonging to the genus *Porphyomonas*, *Tannerella*, *Treponema*, *Fusobatcerium*, etc., were higher in NPC patients' nasopharynx (Fig. 1e). Together, these findings indicated that the microbial niches specialization between the nasopharynx and oral cavity tend to be lost in NPC patients.

Next, to determine whether abnormal oral-to-nasopharyngeal microbial translocation existed in NPC patients, we performed source tracking analysis using the FEAST and SourceTracker2 algorithms on paired oral-nasopharyngeal microbiota data. The outputs of these algorithms represented the contributions of oral microbiota to the nasopharyngeal community, which were defined as "translocation score". We observed higher translocation scores in NPC patients than controls (Supplementary Fig. 5). Next, individuals were distinctly categorized into high/low-translocation groups based on the translocation scores from FEAST and SourceTracker2 algorithms using k-means clustering (Fig. 1f). Interestingly, we observed high-translocation individuals exhibited 4.51 times more risk of NPC when compared with the low-translocation ones (Logistic regression, OR (95%CI) = 4.51 (1.47–16.04), $P = 0.012$, Table 1). The species associated with the translocation were further identified in nasopharynx. A total of 33 species were significantly enriched in the high-translocation group, including five taxa from genus *Prevotella*, four taxa from genus *Streptococcus* and four taxa from genus *Neisseria*, which were dominant in the oral cavity (Fig. 1g). An additional 11 species were significantly depleted in the high-translocation group, they were nasopharyngeal commensals from genus *Corynebacterium*, *Cutibacterium*, *Staphylococcus* and *Dolosigranulum* (including four, three, three and one taxa, respectively, Fig. 1g). Our results revealed that abnormal inflow of oral microbiota into nasopharynx is an important risk factor for NPC.

### Microbial signatures associated with NPC and translocation phenomenon were highly consistent

When we explored the NPC-specific nasopharyngeal microbial characteristics, we found that the microbial communities of NPC patients were obviously different from those of controls (Bray-Curtis distance, PERMANOVA, $R^2 = 0.029$, $P < 0.001$, Fig. 2a). To further seek the NPC-associated features, ANCOM-BC was performed to identify the differentially enriched taxa between NPC patients and controls. A total of 20 taxa that differed significantly in abundance were identified between NPC patients and controls at the species-level (Fig. 2b, c and Supplementary Data 1). Of these, 15 species were enriched in NPC patients. Notably, there was a subset of well-documented oral pathobionts, such as *Prevotella intermedia*, *Fusobacterium nucleatum* and *Peptostreptococcus stomatis*. Five species were depleted in NPC, which were nasopharyngeal commensals dominated the healthy nasopharynx, such as *Corynebacterium accolens*, *Staphylococcus epidermidis* and *Cutibacterium acnes*.

We also observed the distinct patterns of microbial co-occurrence networks between NPC and control group (Fig. 2d, e and Supplementary Data 2, 3). In the NPC group, all 15 NPC-enriched species (pink cycles) were engaged in a positive subnetwork with 26 significant interactions. Among them, *Prevotella intermedia*, *Fusobacterium*

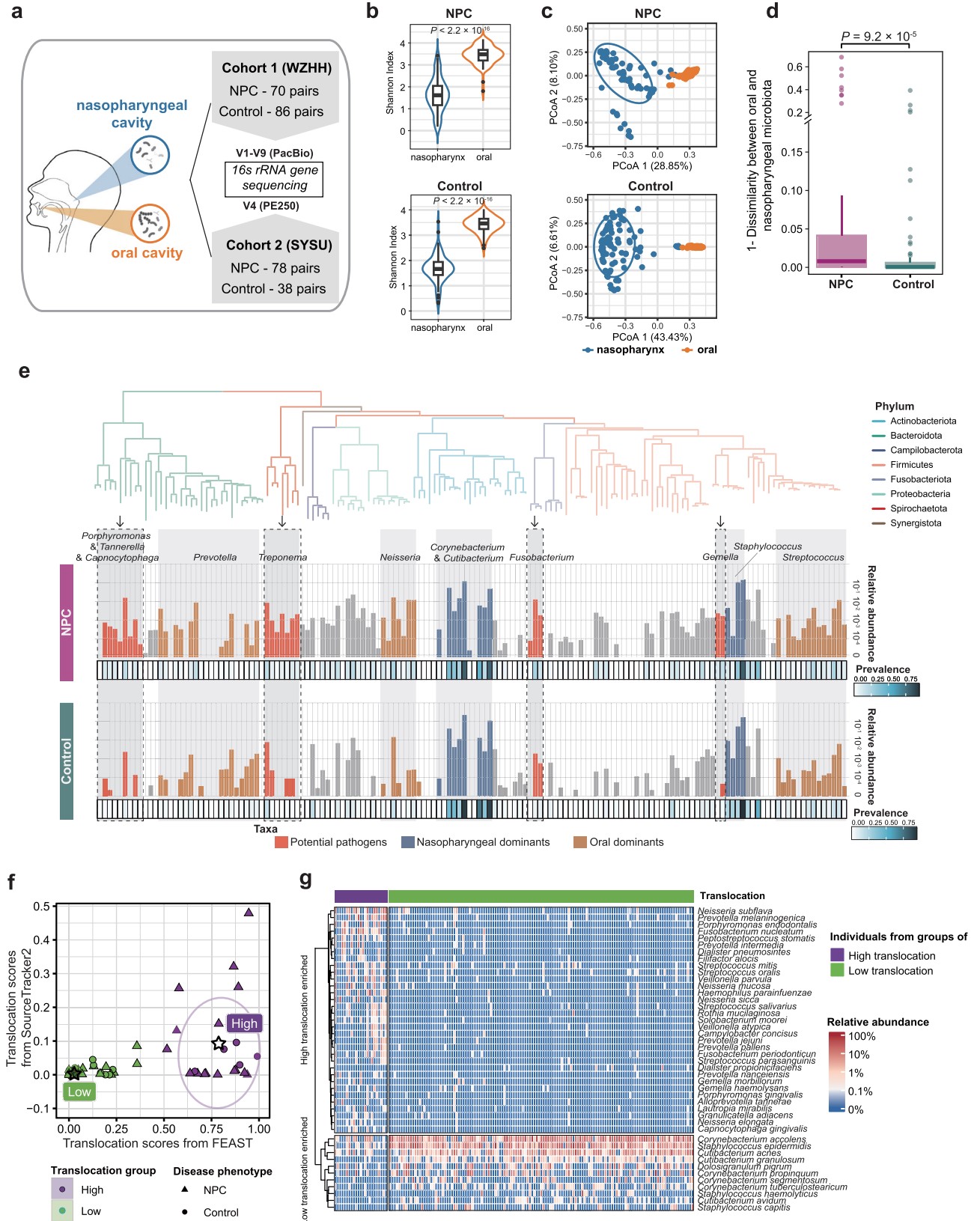

*nucleatum* and *Peptostreptococcus stomatis* occupied the core of the subnetwork with the strongest positive correlations (SparCC, R: 0.439–0.547, Fig. 2d and Supplementary Data 2). But this phenomenon was absent in the control group, where only two NPC-enriched species slightly connected with each other (Fig. 2e and Supplementary Data 3). Meanwhile, we noted the subnetwork, composed of commensals, was

significantly negatively correlated with the subnetwork dominated by oral pathobionts in both groups. The above data suggested NPC-enriched microbes, especially oral pathobionts, were likely to co-occur and reshaped the nasopharyngeal ecology in NPC patients.

We next explore the microbes that were associated with both NPC-risk and oral-to-nasopharyngeal translocation. The results

**Fig. 1 | The influx of oral microbes shaped two types of nasopharyngeal microbial communities. a** Designs of NPC case-control microbiota study. **b** The Violin plots of Shannon alpha diversity index between nasopharyngeal and oral microbiota. *P* values between the two groups were determined by the Wilcoxon rank-sum test (two-sided). Overlayed boxplots were presented with the median marked by thick black line, the interquartile range marked by the white bar, the range by the thin black line and outliners by the black dots ($N = 70$ for NPC, and $N = 86$ for control). **c** The PCoA plots based on Bray-Curtis distance within NPC patients ($N = 70$) or controls ($N = 86$). PCoA analysis was applied in combined NPC and control samples, and plots were presented separately on the same set of axes. Ellipses with 75% levels were shown. **d** The oral-nasopharyngeal paired Bray-Curtis distance between NPC patients and controls. *P* values between the two groups were determined by the Wilcoxon rank-sum test (two-sided). Boxplots were presented with the median marked by thick line, the interquartile range marked by the bar, the range by the thin line and outliners by dots ($N = 70$ for NPC, and $N = 86$ for control). **e** The phylogenetic tree represented ASVs of core species with >5% presence in nasopharyngeal microbiota. The phylogenetic tree was constructed by Mega7 software using the neighbor-joining method. The bar plots indicated the relative abundance of these species in nasopharynx which the filled colors indicate the types of microbes (top three commensal genera of nasopharynx and oral cavity, or potential pathogens). The heatmap indicated the prevalence of microbes in nasopharynx. **f** The scatter plot showing the two translocation clusters divided based on results from FEAST and SourceTracker2 algorithms using k-means methods. Ellipses with 75% levels were shown. Star represented the center of each cluster. **g** The heatmap showing the significant differential species between high- ($N = 21$) /low- ($N = 135$) translocation groups based on nasopharyngeal microbiota data (ANCOM-BC, two-sided, FDR $q < 0.05$) with adjusting for age, sex, cigarette smoking status, alcohol drinking status, whether have caries and whether have oral or nasal diseases. **b–g** were plotted based on data from Cohort 1. PCoA principal coordinate analysis, ASV amplicon sequence variant, FEAST fast expectation-maximization microbial source tracking, ANCOM-BC analysis of compositions of microbiomes with bias correction, FDR false discovery rate. Source data are provided as a Source Data file.

**Table 1 | Odds ratio (OR) estimates of nasopharyngeal carcinoma according to translocation groups**

| Translocation group | NPC (N) | Control (N) | Unadjusted model | | | Adjusted model[b] | | |
|---|---|---|---|---|---|---|---|---|
| | | | OR[a] | 95% CI | *P* | OR[a] | 95% CI | *P* |
| Low ($N = 135$) | 54 | 81 | 1.00 | | | 1.00 | | |
| High ($N = 21$) | 16 | 5 | 4.80 | 1.76–15.38 | $3.78 \times 10^{-3}$ | 4.51 | 1.47–16.04 | $1.22 \times 10^{-2}$ |

[a]ORs and *P* values obtained from Logistic regression model (two-sided).
[b]Adjusting factors were included age, sex, cigarette smoking status, alcohol drinking status, whether have caries and whether have oral/nasal diseases.

showed 13 species were closely associated with both NPC and high translocation (Fig. 2f), including *Fusobacterium nucleatum*, *Prevotella intermedia*, *Peptostreptococcus stomatis*, etc. We defined this subset of species as the "NPC$_{OtoNP}$" microbes. In nasopharynx, NPC patients with high-translocation (NPC_H) harbored a higher abundance of NPC$_{OtoNP}$ microbes. While NPC patients with low-translocation (NPC_L) and controls showed distinct distribution patterns, where these microbes were less observed (Fig. 2g). We noticed that the differences of nasopharyngeal communities between NPC_H and controls (PERMANOVA, $R^2 = 0.093$) were 5.17 times greater than that between NPC_L and controls (PERMANOVA, $R^2 = 0.018$) (Fig. 2h). By constructing random forest models using "NPC$_{OtoNP}$" microbes as features, we observed that the model welly discriminated NPC_H and control groups with AUC of 0.926 (95% CI: 0.847–1.000), showcasing that the "NPC$_{OtoNP}$" microbes were closely correlated to NPC risk. While the models' performance in NPC_L *vs* control groups and total NPC *vs* control groups were unsatisfactory with much lower accuracy (AUCs of 0.708 (95%CI: 0.632–0.784) and 0.745(95%CI: 0.678–0.812)) (Fig. 2i). These results collectively suggested that the existence of translocated oral pathobionts in the nasopharynx was highly correlated to NPC risks.

### Culturomics validated the translocation of *Fusobacterium nucleatum* and *Prevotella intermedia* in NPC patients

To validate the presence of oral-to-nasopharyngeal translocation in NPC patients, we examined whether the identical strains could be isolated from nasopharynx and oral cavity of the same individual using the culturomics approach described in Fig. 3a. In total, paired nasopharyngeal swabs and saliva specimens were prospectively collected from 48 individuals, including 34 NPC patients and 14 non-tumor controls (Fig. 3b and Supplementary Data 4). In the pilot culturomics test with Columbia blood agar, the nasopharyngeal isolates were almost facultative anaerobic local symbioses and opportunistic pathogens, mainly *Staphylococcus* and *Klebsiella*. Considering the biological profiles of translocated microbes, selective culturomics with kanamycin-vancomycin was performed. Colonies were successfully cultured from five nasopharyngeal and all 48 oral specimens. Of note, all colony-positive nasopharyngeal samples were obtained from NPC

patients (Supplementary Data 4). We identified taxonomic information of 541 colonies isolated of nasopharyngeal and oral samples from these five NPC patients with colonies in nasopharynx, which mainly from genus *Prevotella*, *Fusobacterium* and *Bacteroides*. "NPC$_{OtoNP}$" microbes, concretely, *Fusobacterium nucleatum* and *Prevotella intermedia* were detected in paired two-site specimens from two and three patients, respectively. Notably, we isolated multiple species belonging to genus *Prevotella* in oral cavity, but only *Prevotella intermedia* was isolated from the nasopharynx (Fig. 3d and Supplementary Data 4), implying a possible stronger translocation capacity of *Prevotella intermedia* rather than other *Prevotella spp.*

Further, to confirm the oral-to-nasopharyngeal microbial translocation in NPC patients at the strain level, we used AP-PCR as strain classifying method to identify the consistency of the bacteria isolated from two sites of the same patient. We isolated two *Fusobacterium nucleatum* strains in both nasopharyngeal and saliva specimens of the same patient S007, and another strain was isolated from the nasopharynx of patient S038, which was also present in the corresponding oral cavity (Fig. 3c). For *Prevotella intermedia* isolates, each of three patients (S001, S024 and S027) had a specific strain detected in the nasopharynx as well as in saliva (Fig. 3d). Additionally, we conducted whole-genome sequencing on the isolated strains and observed an average nucleotide identity (ANI) of 99.999% between isolates from the same individual's nasopharyngeal and oral samples (Fig. 3e, f). In contrast, the inter-individual ANI for *Fusobacterium nucleatum* and *Prevotella intermedia* was approximately 95% and 97%, respectively. The phylogenetic analysis also suggested that isolates within the same individual had closer genetic evolutionary distances (Fig. 3g, h). These findings indicated that the nasopharyngeal and oral cavity of NPC patients shared identical strains and confirmed the phenomenon of microbial translocation by the culture-based method.

### Oral translocated microbes infiltrated intra tumor and altered tumor microenvironment

We next asked whether oral pathobionts translocated to nasopharynx could colonize intratumor and participate in tumor microenvironment reconstruction. Using fluorescence in situ hybridization (FISH), we

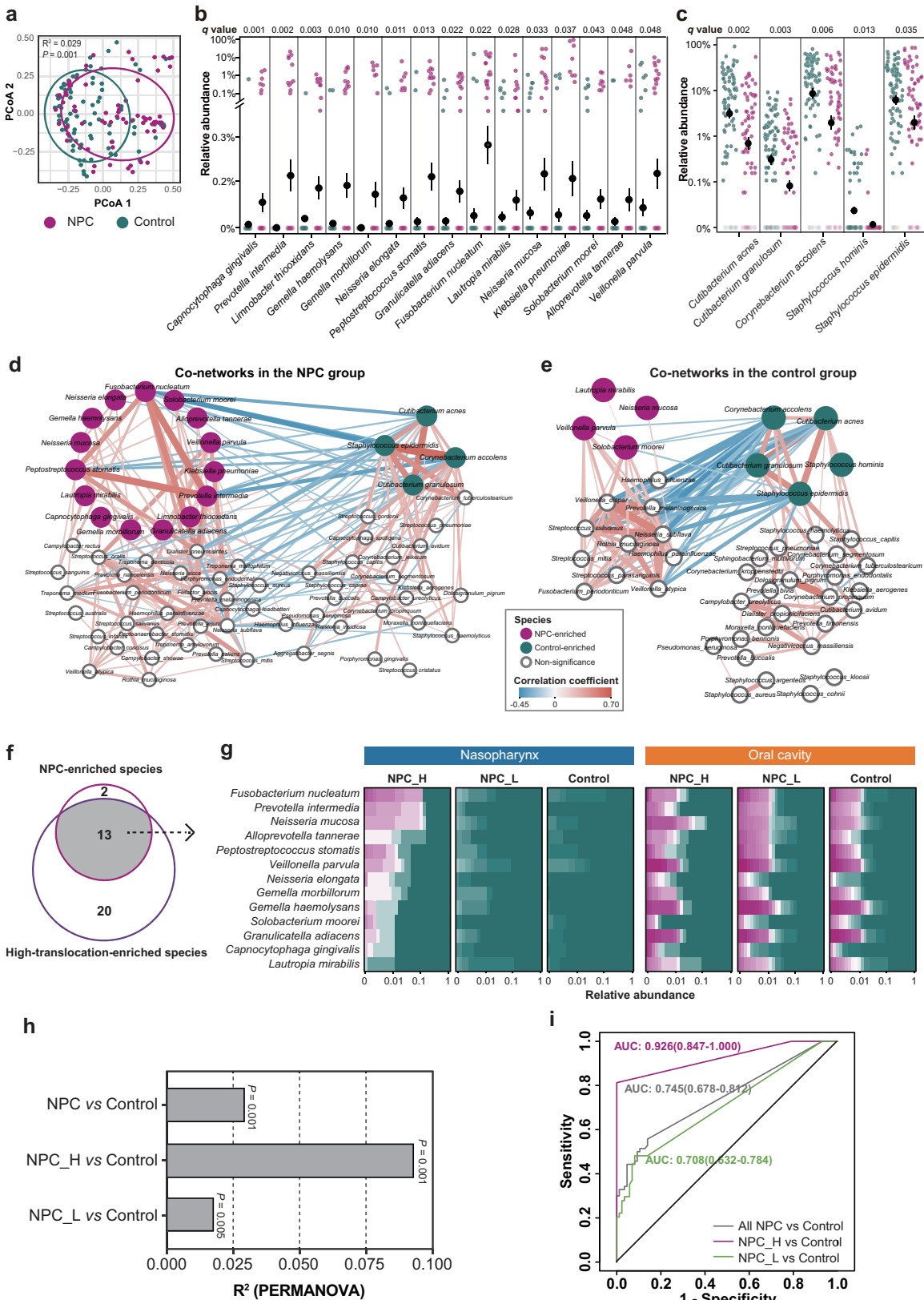

verified the presence of two representative "NPC$_{OtoNP}$" microbes, *Fusobacterium nucleatum* and *Prevotella intermedia*, in NPC tumors (Fig. 4a). To further elucidate the microbiota profiles and their interaction with the microenvironment in nasopharyngeal tissues, we identified microbial features and correlated host alteration using meta-transcriptomic data from 89 NPC tumoral and 12 normal tissues. The

listing of species and their prevalence of the NPC tumors and normal tissues was shown in Supplementary Data 5. Interestingly, nine of 13 "NPC$_{OtoNP}$" microbes were indeed observed intra tumors (Fig. 4b). Of these, *Fusobacterium nucleatum*, the most prevalent one, could be detected in 29.2% of NPC patients but only in 8.3% of controls, and *Prevotella intermedia*, the second most prevalent one, was only

**Fig. 2 | Characteristic microbial communities appeared in the nasopharynx of NPC patients. a** The PCoA plot based on Bray-Curtis dissimilarity distance showing the compositional differences between NPC ($N = 70$) and control ($N = 86$) groups with adjusting for the confounding variables ($R^2$ and $P$ value was obtained from PERMANOVA, two-sided). The dot plots showing the relative abundances of NPC-enriched (**b**) and NPC-delepted (**c**) species (ANCOM-BC, two-sided, FDR $q < 0.05$) with adjusting for the confounding variables. Data represented mean ± SEM ($N = 70$ for NPC, and $N = 86$ for control). The co-occurrence network of nasopharyngeal microbiota in NPC patients (**d**, $N = 70$) and controls (**e**, $N = 86$). Only significant correlations were shown in the networks (SparCC, $|r| > 0.25$ and $P < 0.05$, two-sided). Each node represented a microbial species, NPC-enriched and control-enriched species were shown in pink and green, respectively. Each edge represented the correlation between paired species, and its width reflected the absolute value of the correlation coefficient. Co-inclusion associations were colored in red, whereas co-exclusion correlations were colored in blue. **f** The venn plot showing 13 species were both NPC-enriched and high-translocation-enriched which were

defined as "NPC$_{OtoNP}$" species. **g** The heatmaps show the relative abundance and prevalence of "NPC$_{OtoNP}$" species in oral and nasopharyngeal samples from NPC patients ($N = 70$) and controls ($N = 86$). **h** The bar plots showing the $R^2$ and $P$ values obtained from PERMANOVA (two-sided) of NPC ($N = 70$)/NPC_H ($N = 16$)/NPC_L ($N = 54$) vs control ($N = 86$) groups with 1001 permutations. **i** The ROC curves of all NPC vs control, NPC_H vs control, and NPC_L vs control groups, corresponding AUCs (95%CI) were shown. Data were based on the nasopharyngeal microbiota of Cohort 1. The confounding variables contain age, sex, cigarette smoking status, alcohol drinking status, whether have caries and whether have oral or nasal diseases. PCoA principal coordinate analysis, PERMANOVA Permutational multivariate analysis of variance, ANCOM-BC analysis of compositions of microbiomes with bias correction, FDR false discovery rate, SparCC sparse correlations for compositional data algorithm, NPC_H NPC patients with high-translocation, NPC_L NPC patients with low-translocation, ROC Receiver operating characteristic, AUC Area Under the Curve, 95% CI 95% confidence intervals. Source data are provided as a Source Data file.

detectable in tumor tissues but not in controls, with a detection rate of 10.1%. These together indicated the existence of translocated oral microbes in the tumor microenvironment.

To reveal the influence of these translocated microbes on the tumor microenvironment, we classified NPC patients into two groups according to the presence/absence of these microbes in tissues: OtoNP$^+$ ($N = 29$) and OtoNP$^-$ ($N = 60$). Using immune cell enrichment analysis by ssGSEA, we found that neutrophils were specifically enriched in OtoNP$^+$ tumors other than OtoNP$^-$ tumor and normal tissues (Fig. 4c). We identified that 103 genes were upregulated, whereas 47 genes were downregulated in the OtoNP$^+$ tumors than OtoNP$^-$ one (FDR-$q < 0.05$, $|\log_2 FC| > 1$, Fig. 4d and Supplementary Data 6). The enrichment analysis of Gene Ontology (GO) biological processes between OtoNP$^+$/OtoNP$^-$ groups showed significant enrichment of the migration or chemotaxis of myeloid cells, especially neutrophils (FDR-$q < 0.05$, Fig. 4e and Supplementary Data 7). Interestingly, several representative genes of tumor-associated neutrophils were highly expressed in OtoNP$^+$ group, including *OSM* and *TREM1* (Fig. 4d and Supplementary Data 6), which further suggest that neutrophils might participate in shaping the tumor immune microenvironment by intratumoral translocated microbes. Kyoto Encyclopedia of Genes and Genomes (KEGG) pathway analysis recognized 10 significantly enriched pathways (Fig. 4f and Supplementary Data 8). Of these, the most significant one was the viral protein interaction with cytokine and cytokine receptor pathway. Considering EBV infection is an important pathogen of NPC, the enriched viral-associated pathway implied that there might be synergistic interactions between these intratumoral microbes and EBV infection.

**Translocation-related microbes influenced the EBV infection in nasopharynx**

Results from the above meta-transcriptomic data provided evidence for a potential link between the presence of "NPC$_{OtoNP}$" microbes and viral infection. Given the key roles of EBV in the pathogenesis of NPC, we investigated the association between nasopharyngeal microbiota and EBV infection. We evaluated the EBV load in nasopharyngeal environment and classified individuals into four groups of non-detected, low, medium and high groups accordingly ($N = 78, 25, 25, 25$, respectively). We observed that with the higher degree of EBV load, the numbers of detected translocated species were significantly increased (Fig. 5a, b), this association remained significant when analyzing the NPC patients and controls separately (Supplementary Fig. 6). Among the 80 core nasopharyngeal microbes (with a detection rate > 5%), we identified six species whose abundances were significantly correlated to EBV loads (Spearman correlation, FDR-$q < 0.1$, Supplementary Data 9). The six positively correlated species were all "NPC$_{OtoNP}$" species (Fig. 5c). *Fusobacterium nucleatum* was observed with the most significant positive correlation with nasopharyngeal EBV load

(Spearman correlation, $R = 0.36$, $P = 4.2 \times 10^{-6}$, Fig. 5c). Other significant species were included *Peptostreptococcus stomatis*, *Prevotella intermedia*, *Gemella morbillorum*, *Capnocytophaga gingivalis* and *Gemella haemolysans* (Fig. 5c and Supplementary Data 9). Four species that showed a negative correlation to EBV loads were the taxa significantly depleted in NPC patients, including *Corynebacterium accolens*, *Cutibacterium granulosum*, *Cutibacterium acnes* and *Staphylococcus epidermidis* (Fig. 5d). Moreover, we observed the stable protection effects of these nasopharyngeal commensals on the oral translocated microbes. The higher abundance of nasopharyngeal commensals, the lower number of oral translocated microbes detected (Supplementary Fig. 7). The above findings indicated the abnormal nasopharyngeal microbiota alterations were closely associated with the EBV infection in the nasopharynx.

## Discussion

Here, we investigated the alterations of nasopharyngeal microbiota for NPC patients, the oral-to-nasopharyngeal microbial translocation and its association with NPC pathogenesis. Utilizing paired nasopharyngeal swabs and saliva samples, we revealed an unusually oral-to-nasopharyngeal microbial translocation, which was strongly associated with increased NPC risk. These oral-derived microbes, represented by *Fusobacterium nucleatum* and *Prevotella intermedia*, were significantly enriched in the nasopharynx and reshaped the nasopharyngeal microbiota of NPC patients. These translocated microbes were involved in intratumoral infiltration and tumor microenvironment remodeling and were closely linked to epithelial EBV infection, which might play important roles in NPC pathogenesis.

One of our key findings was the loss of ecological specialization between nasopharynx and oral cavity in NPC patients, which was driven by the translocation of specific oral taxa to the nasopharynx. And we observed that the NPC risks increased with a higher influx of oral bacteria into nasopharynx, suggesting the important roles of oral-to-nasopharyngeal translocation in NPC. As reported, the influences of oral microbiota on human health extend beyond the oral cavity. Many systemic conditions such as colorectal carcinoma and inflammatory bowel disease are closely associated with oral microbiota, especially the translocated oral microbes[5,6], illustrating the importance of maintaining ecological topography and the pathogenicity of oral pathobionts in the extraoral niches. Previous studies also showed that loss of oral-nasopharyngeal topography was related to the higher susceptibility to respiratory infections in infants and the elderly[10,16], but evidence on carcinogenesis is still lacking. Since the nasopharynx is anatomically connected to the oral cavity, which provides opportunities for microbial communication between these two sites. Our study revealed that oral pathobionts abnormally trans-colonized in nasopharynx and reshaped the local microenvironment, which was associated with NPC. Future studies are needed to explore the

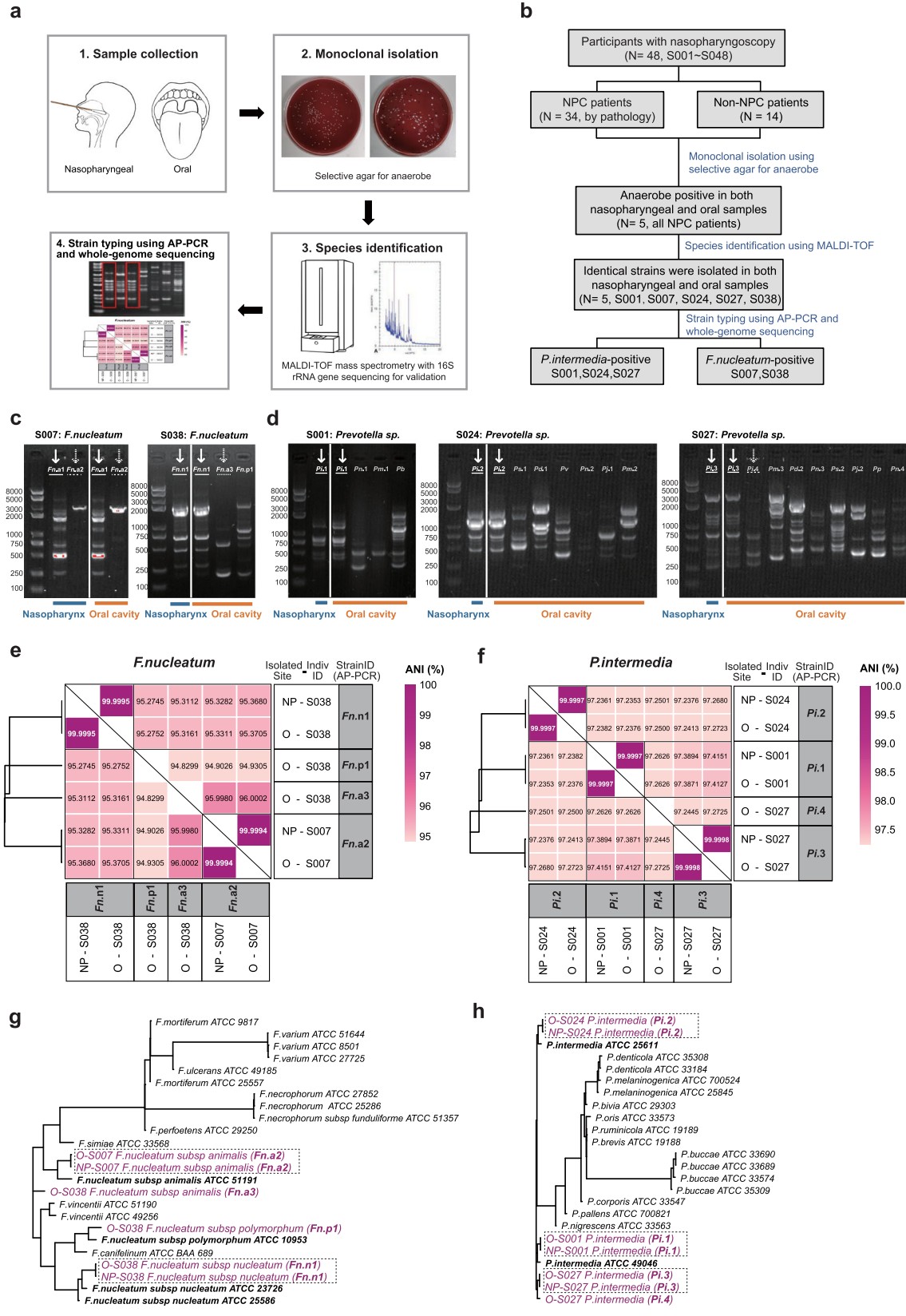

phenomenon and significance of bidirectional microbial translocation between these two sites in nasopharyngeal and oral diseases.

Our results showed the loss of nasopharyngeal microbial specialization in NPC patients also covaried with the loss of putative beneficial commensal taxa in the nasopharynx, as reflected by significant depletion of *Corynebacterium accolens*, *Staphylococcus epidermidis*,

*Staphylococcus hominis*, *Cutibacterium acnes* and *Cutibacterium granulosum* in NPC patients. These bacteria reside in the normal nasopharynx of adults and might participate in the maturation and maintenance of respiratory physiology and immune homeostasis[17]. "Colonization resistance", defending against pathogen colonization and infection, is also considered an important physiological function

**Fig. 3 | Identical microbial strains colonized in the nasopharynx and oral cavity of NPC patients. a** The schematic of the culturomics design. Nasopharyngeal swabs and saliva samples were collected and cultured in anaerobic conditions; clones were identified by MALDI-TOF MS combined with Sanger sequencing of 16S rRNA genes. Typical clones were performed strain typing using AP-PCR and whole genome sequencing. **b** The flowchart of the study. AP-PCR patterns of *Fusobacterium nucleatum* from S007 and S038 (**c**) and *Prevotella intermedia* from S001, S024 and S027 (**d**). Identical strains were highlighted with line/dashed lines. Three experiments were repeated independently with similar results. The comparison of ANI between different isolates of *Fusobacterium nucleatum* (**e**) and *Prevotella*

*intermedia* (**f**). **g, h** Phylogenetic trees for genus *Fusobacterium* and *Prevotella*. MALDI-TOF MS, Matrix Assisted Laser Desorption/ionization Time-Of-Flight mass spectrometry, AP-PCR arbitrarily primed polymerase chain reaction, ANI average nucleotide identity, *F.nuleatum Fusobacterium nucleatum*, *P.intermedia Prevotella intermedia*, *Fn.a Fusobacterium nucleatum subsp. Animals*; *Fn.n, Fusobacterium nucleatum subsp. nucleatum*; *Fn.p, Fusobacterium nucleatum subsp. polymorphum*; *Pi, Prevotella intermedia*, *Pm, Prevotella melaninogenica*; *Ps, Prevotella salivae*; *Pd, Prevotella denticola*; *Pn, Prevotella nigrescens*; *Pj, Prevotella jejuni*; *Pp, Prevotella pallens*. Source data are provided as a Source Data file.

for these commensal bacteria. For example, the *Staphylococcus epidermidis* colonized in the nasal cavity of healthy people could protect the respiratory epithelial cells from influenza virus infection by inducing the production of interferon[18]. Moreover, *Corynebacterium pseudodiphtheriticum*, another respiratory commensal bacterium of *Corynebacterium*, could improve the resistance of mice to the respiratory syncytial virus and secondary pneumonia[19]. Of note, a nasal *Corynebacterium pseudodiphtheriticum* probiotic preparation was shown the safety and effectiveness in eliminating *Staphylococcus aureus* infection in a small pilot study[20]. Therefore, these commensal bacteria have the potential to be exploited as probiotics, and their usage in the upper respiratory tract might protect against virus infection and prevent the ectopic colonization of oral microbes.

Intratumoral bacteria have been detected in multiple human cancers, including those previously thought to be sterile[21]. Qiao et al. and Zhong et al. demonstrated the phenomenon that bacteria were present in NPC tumors and correlated with the clinical prognosis of NPC patients[13,15]. Notably, several taxa associated with the worse prognosis of NPC patients were predominantly oral residents, such as *Prevotella* and *Porphyromonas*. While limited to research methods or relatively small populations, previous studies did not focus on what species inside the tumor, especially the oral pathobionts at low abundance and prevalence. Here we showed the presence of microecology in nasopharyngeal tissues using the meta-transcriptomic data and confirmed the translocation of "NPC_OtoNP" microbes into NPC tumors with the FISH analysis.

Intratumoral microbes could influence the biological process of human cancers in several ways, including producing virulence factors that affect cell signal transduction and regulating host local and systemic immunity[22]. Chronic inflammation has been recognized as a hallmark of tumors, which could often be induced by locally aberrant microbes[23]. In immune cell enrichment analysis, we noted that the neutrophils were specifically enriched in OtoNP+ NPC tumors, accompanied by the upregulation of the expression of representative signaling *CXCL8-CXCR1/2* axis. Neutrophils are the first line of human immune system to defend against infection, and growing evidence indicates that they play an important role in the initiation and progression of tumors[24,25]. In most tumors, including NPC, the high infiltration of neutrophils is strongly associated with poor clinical prognosis[26,27]. The interactions between microbiota and neutrophils indicated in this study might impact tumor progression. For example, *Porphyromonas gingivalis* has been reported to shape a neutrophil-predominant tumor microenvironment via the increased secretion of neutrophilic chemokines and neutrophil elastase to promote the progression of pancreatic carcinoma in mice[28]. In addition, multiple therapeutic approaches targeting neutrophils have emerged, with several reagents entering clinical trials, such as the *CXCR1/2* inhibitor SX-682 for metastatic melanoma (NTC03161431), from which OtoNP+ NPC patients might benefit. Thus, further studies to better understand the possible effects of intratumoral bacteria on the tumor microenvironment might be beneficial to explore new therapeutic options for patients with cancer.

Persistent EBV infection in epithelial cells is considered the essential step for initiating the tumorigenic transformation of NPC.

While chronic exposure of the nasopharyngeal epitheliums to environmental carcinogens, like salted fish and tobacco, is believed to increase the risk of infecting and establishing latency of EBV in epitheliums[11], the factors that promote epitheliums establishing viral latency remain poorly understood. Current understanding about whether and how the microbiota participates in host EBV infection, especially in the latent infection in the epitheliums, is very insufficient. One study reported that the adhesion of *Helicobacter pylori* induced the expression of accessory EBV receptors in gastric epithelial cells and increased the efficiency of EBV infection[29]. Another earlier study also reported that the culture fluid of *Fusobacterium nucleatum* could efficiently induce EBV-associated antigens in vitro[30]. Besides, epidemiological observations revealed the EBV and oral pathogens co-infections and their association with worse progression of oral diseases[31], indicating the pathogenic effects of EBV-bacterial synergies. In this study, we observed the abundance of oral-translocated microbes in nasopharynx was significantly positive-correlated to local EBV burden. Combining the evidence of viral-associated microenvironment alteration from intratumoral meta-transcriptomic data, we put forward a possible hypothesis that translocated oral microbiota may act synergistically with EBV in the pathogenesis of NPC, which together enable favorable conditions for latent infection. Constrained by the cross-sectional study design, our study can only provide preliminary evidence of the correlation between nasopharyngeal microbiota and EBV load. To substantiate these findings, further prospective studies with larger sample sizes are warranted.

The translocation of oral microbes to extra-oral sites has been gradually observed in several cancers. Studies reported that translocated oral microbes have been present in precancerous lesions and involved in precancerous-carcinoma processes[32-34]. For example, *Fusobacterium nucleatum*, were significantly enriched in colorectal adenomas[35], whose gut colonization could directly promote cell proliferation through the activating various signaling pathways such as Wnt/β-catenin, or shape the pro-tumor microenvironment by regulating the secretion of pro-inflammatory cytokines indirectly[36]. In addition, immunosuppressive microenvironment of the tumors might also facilitate the colonization of certain microbes[37]. In this study, we could not determine whether the migration of microbes promotes the onset of NPC, or if the altered tumor microenvironment facilitates the colonization of specific oral microbes. Further prospective cohort studies and functional experiments are necessary to address the roles of microbial translocation in the NPC pathogenesis.

There are some limitations in this study. Although PacBio full-length 16S rRNA sequencing method provided us with high accuracy and species-level taxonomic resolution, allowing cost-effectively conduct taxonomic exploration of samples with high host-DNA background (such as nasopharyngeal swabs), further technological innovations to increase the feasibility of metagenomic sequencing would be more conducive to functional research of microbial communities. Additionally, the utilization of both 16S rRNA gene full-length and V4 region sequencing methodologies presents a hurdle in combining data for species identification. In light of this, we advocate for the adoption of consistent sequencing methods across studies to facilitate seamless data integration and validation analyses across

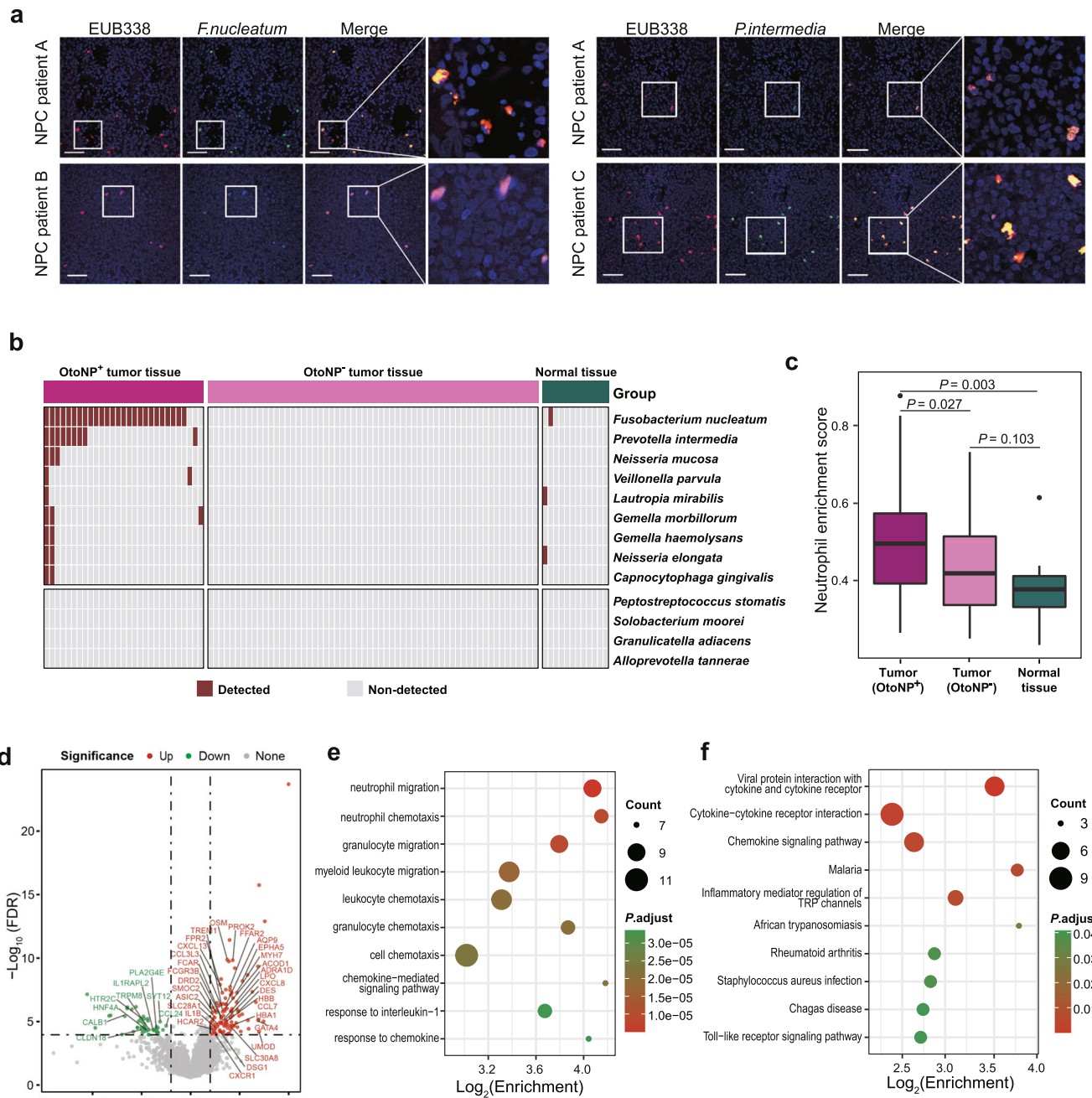

**Fig. 4 | Oral translocated microbes infiltrated the tumor and affected the tumor microenvironment. a** Representative images of the presence of *Fusobacterium nucleatum* (left) and *Prevotella intermedia* (right) in tumor tissues of NPC patients. EUB338 (red) was a Cy3-conjugated universal bacterial oligonucleotide probe. *F. nucleatum* and *P. intermedia* were FITC-conjugated oligonucleotide probes (green). High-magnification images of the boxed area are shown on the right. Scale bars, 50 µm. Three and four patients were identified for *F. nucleatum* and *P. intermedia* with similar results. **b** The detection status of "NPC_{OtoNP}" microbes in metatranscriptomic sequencing data of nasopharyngeal tissues ($N = 29$ for OtoNP$^+$ tumors, $N = 60$ for OtoNP$^-$ tumors and $N = 12$ for normal tissues). **c** The boxplot showing the enrichment score for neutrophil among OtoNP$^+$ tumors ($N = 29$), OtoNP$^-$ tumors ($N = 60$) and normal tissues ($N = 12$). *P* values were determined by *t* test (two-sided). Boxplots were presented with the median marked by thick black

line, the interquartile range marked by the bar, the range by the thin line and outliners by the black dots (**d**) The volcano plot showing the differentially expressed genes between OtoNP$^+$ ($N = 29$) and OtoNP$^-$ ($N = 60$) tumor tissues (edgeR, two-sided, FDR-$q < 0.05$ & |log$_2$FC| > 1). Genes involved in top 10 significant GO and KEGG pathways were labeled in the plot. The results of pathways enrichment analysis of differential genes based on GO (**e**) and KEGG pathways (**f**). "NPC_{OtoNP}" microbes, the species that NPC-enriched and high-translocation-enriched; All pathways were with the FDR-adjusted $P < 0.05$ (two-sided). OtoNP$^+$ tumor, the tumor with "NPC_{OtoNP}" microbes; OtoNP$^-$ tumor, the tumor without "NPC_{OtoNP}" microbes, FISH fluorescence in situ hybridization probes, ssGSEA single sample Gene Set Enrichment Analysis, GO Gene Ontology, KEGG Kyoto Encyclopedia of Genes and Genomes, FDR false discovery rate. Source data are provided as a Source Data file.

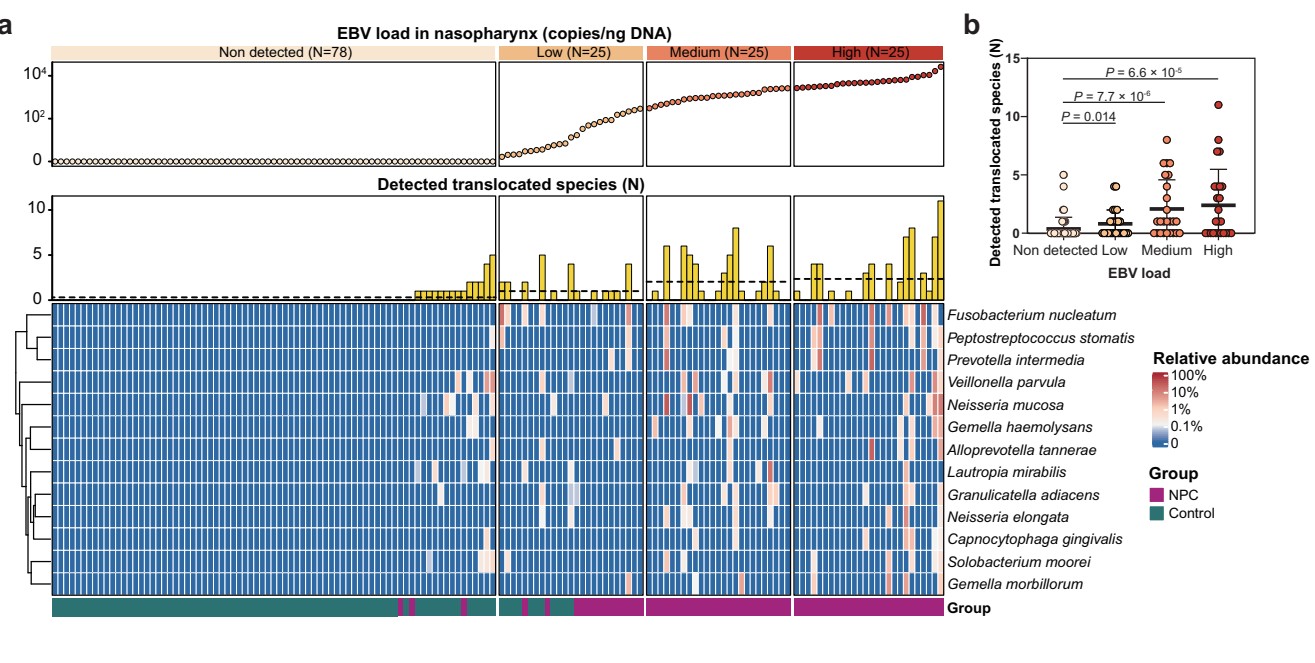

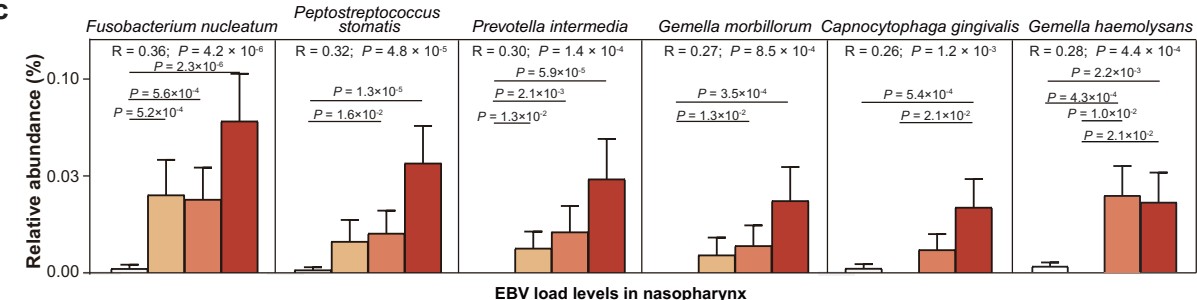

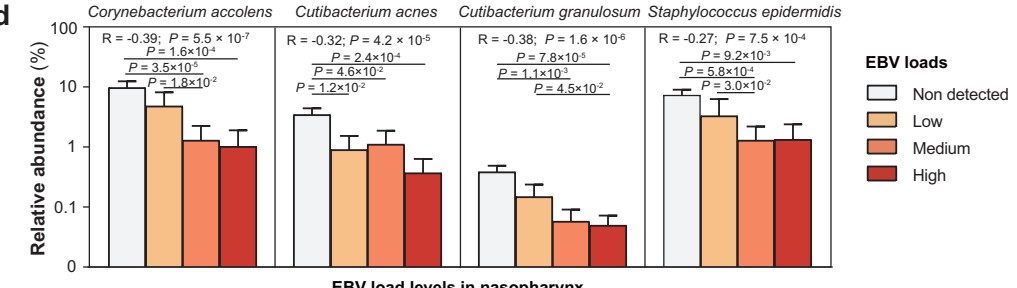

**Fig. 5 | The association between oral translocated microbes and EBV infection in nasopharynx. a** Correlation between EBV loads and translocated ("NPC$_{OtoNP}$") species in nasopharynx. Individuals were classified into four groups according to EBV load: non-detected group (EBV were not detected in samples, $N = 78$), and low, medium and high groups (for samples with EBV detected, individuals were divided by 33% and 67% quartiles, $N = 25, 25, 25$ representatively). Each column represented an individual. EBV loads were shown by dot plots on the top; the corresponding data of the number of detected "NPC$_{OtoNP}$" species were shown by bar plots in the middle; and the relative abundances of "NPC$_{OtoNP}$" species were shown by heatmap at the bottom. **b** The association between the number of detected "NPC$_{OtoNP}$" microbes and EBV loads in the nasopharynx ($N = 153$). $P$ values were determined by

the Wilcoxon rank-sum test (two-sided). **c** The correlation analyses between "NPC$_{OtoNP}$" microbes and EBV load were performed using Spearman's correlation coefficient for ranked data, and $P$ values between the two groups were determined by the Wilcoxon rank-sum test (two-sided, $N = 153$). **d** The correlation analyses between control-enriched microbes and EBV load were performed using Spearman's correlation coefficient for ranked data, and $P$ values between the two groups were determined by the Wilcoxon rank-sum test (two-sided, $N = 153$). Data was shown in mean ± SEM. NPC nasopharyngeal carcinoma, EBV Epstein-Barr virus, "NPC$_{OtoNP}$" microbes, the species that NPC-enriched and high-translocation-enriched. Source data are provided as a Source Data file.

diverse cohorts. Other limitations included the relatively low sequencing depth of meta-transcriptomic data after filtering host reads, which limited our ability to perform accurate taxonomic abundance estimates and explore the microbial transcriptional features and related functional pathways. Also, the meta-transcriptomic data in this study were derived from a limited sample size of normal tissues and were conducted without sampling controls, more rigorous designs should

be applied in future research. Since the nasopharynx is the cavity that contacts not only the oral cavity but also the nasal cavity and lower respiratory tract, its local microbial composition may also be influenced by microbiota from other connecting sites, which is needed for further research.

In conclusion, our study uncovers the oral-to-nasopharyngeal microbial translocation and its potential roles in NPC development,

supporting a new paradigm in the pathogenesis of NPC involving a microbes-EBV synergistic interaction in the local microenvironment of the nasopharynx. Blocking the translocation from the oral cavity to the nasopharynx, such as improving oral hygiene, nasopharyngeal irrigation, or use of probiotics, might become the possible interventions to prevent NPC.

## Methods

This study was approved by the ethics committee of Sun Yat-sen University Cancer Center. Informed consent was obtained from all study participants.

### Study design

The objective of this study was to comprehensively characterize the nasopharyngeal microbiota associated with NPC and to explore the source of potentially pathogenic microbes and their underlying mechanism affecting the occurrence of NPC. Totally, 165 pathologically confirmed, treatment-naïve NPC patients and 138 non-malignant controls from three hospitals were included in this study. All individuals had undergone nasopharyngeal endoscopy. Among them, 84 NPC patients and 94 controls were enrolled in Wuzhou Red Cross Hospital in Wuzhou city, Guangxi province, China, from June 2020 to November 2020. This group served as the discovery cohort (Cohort 1, WZHH) to explore the profiles of NPC-associated nasopharyngeal microbiota at the species-level using Pacbio 16S rRNA gene full-length sequencing. In addition, 81 NPC patients and 44 controls registered in Sun Yat-sen University Cancer Center (SYSUCC) and the First Affiliated Hospital of Sun Yat-sen University in Guangzhou city, Guangdong province, China, from June 2018 to August 2019, were included as the validation cohort (Cohort 2, SYSU) to verify the results at the genus-level using 16S rRNA gene V4-region sequencing. The detailed information including age, sex, cigarette smoking status, alcohol drinking status and oral health-related statuses, were shown in Supplementary Table 1. In addition, for our culturomics study, we prospectively collected paired nasopharyngeal swabs and saliva samples from a separate cohort comprising 48 individuals. These samples were obtained during the nasopharyngoscopy procedures conducted at SYSUCC between May 2022 and June 2022, as well as between July 2023 and August 2023. In this study, we did not have any specific requirements for participants' gender, and the gender was determined through self-reporting.

### Sample collection

The nasopharyngeal swab was obtained with sterile swab under the guidance of endoscopy by an experienced physician. Briefly, the swab was inserted into nasopharynx via nose, rotated several times over mucosal epithelium and withdrawn. Meanwhile, 2.5 mL of unstimulated saliva was also collected using sterile tubes. To avoid possible contamination, non-contact swabs and empty tubes with medium added were collected as sampling controls parallelly. Nasopharyngeal tissues from NPC or other benign diseases were obtained in the previous study[38]. All specimens were stored at −80 °C until required.

### DNA extraction

Total DNA was extracted from swabs and saliva using the DNeasy PowerSoil kit (Qiagen, Cat #: 47014) following the handbook. DNA quality and concentrations were evaluated by Nanodrop 2000 (Invitrogen) and Qubit 3.0 fluorometer (Invitrogen). All sampling controls and distilled water (extraction controls) were used for DNA isolation in the same process.

### Quantitative PCR

The EBV DNA loads in nasopharyngeal swabs were quantified by quantitative PCR (qPCR) toward the BamHI-W region[39]. The sequences of amplified primers were as follows: BamHI-W-F (5′-CCC AAC ACT CCA

CCA CAC C-3′); BamHI-W-R (5′-TCT TAG GAG CTG TCC GAG GG-3′). The sequence of the probe for EBV was FAM-5′-CAC ACA CTA CAC ACA CCC ACC CGT CTC-3′-TAMRA. The gradient-diluted standard samples ($10^7$, $10^6$, $10^5$, $10^4$, $10^3$ and $10^2$ copies/μL) of BamHI-W region were used to obtain the standard curve for absolute quantification. The EBV DNA load in nasopharyngeal swab samples was expressed as copies/ng DNA.

### PacBio full-length 16S rRNA gene sequencing and analysis

The full-length 16S rRNA gene was amplified with universal primers 27F (5′-AGR GTT YGA TYM TGG CTC AG-3′) and 1492R (5′-RGY TAC CTT GTT ACG ACT T-3′) containing 12-bp barcodes. The swabs and saliva were amplified using KAPA HiFi HotStart DNA Polymerase (KAPA Biosystems, Cat #: 7958935001) for 33 and 27 cycles with denaturation at 95 °C for 30 s, annealing at 57 °C for 30 s, and extension at 72 °C for 1 min, respectively. The DNA isolated from sampling and extraction controls as well as distilled water (PCR controls) were also included as PCR templates for amplification. The PCR products were purified with Agencourt AMPure XP (Beckman Coulter, Cat #: A63881) and pooled in equimolar concentration. The SMRTbell libraries were prepared from the purified amplicons by adapter-ligated and sequenced using the PacBio Sequel platform (Pacific Biosciences). The high-quality circular consensus sequence (CCS) reads were obtained from the raw PacBio sequencing data using SMRT Link software (v9.0.0, Pacific Biosciences). Multiplexed libraries were assigned to each sample using Lima (v2.0.0) based on the barcodes. The customized DADA2 (v1.22.0) workflow for PacBio full-length 16S sequencing data was used for quality control, denoising and amplicon sequence variants (ASVs) identification of the demultiplexed CCS[40].

### Illumina 16S rRNA gene sequencing and analysis

The V4 amplicon libraries were constructed by amplifying V4 hypervariable region of 16S rRNA gene was amplified using barcode-labeled primers 515F (5′-GTG CCA GCM GCC GCG GTA A −3′) and 806R (5′-GGA CTA CHV GGG TWT CTA AT-3′) with 32 cycles for swabs and 20 cycles for saliva[41]. Next, the Illumina adapters were added in a secondary 10 cycles of amplification. The cycling conditions were denaturation at 98 °C for 20 s, annealing at 55 °C for 15 s, and extension at 72 °C for 30 s. Simultaneous amplification of negative controls was also performed. The products were separated by agarose gel electrophoresis, then recovered and purified using Agencourt AMPure XP. After quantification using Qubit, equimolar products were pooled and sequenced with the Illumina MiSeq 2500 platform. After demultiplexing the raw data based on barcodes, the de-noising of the sequences was performed by DADA2[42], and the ASVs were finally identified.

### Microbial data analysis based on ASVs

Taxonomic annotation of ASVs was performed against the silva_nr99_v138_train_set sequence database. If possible, the species-level annotation was supplemented using the silva_species_assignment_v138 sequence database. A series of filters were applied to detect and remove the potential contamination following the 'RIDE checklist' (Fig. S1)[43]. In detail, the ASVs deriving from mitochondria or chloroplast, or failing to be annotated at the phylum level of bacteria were removed. Next, potential environmental contamination was identified and filtered by R package Decontam (v1.10.0), combined with the sampling, extraction and PCR controls[44]. In addition, the chimeric and low-depth ASVs were also excluded. For downstream analyses, full-length and V4-region data were rarefied to 2133 and 5000 per sample, respectively, and only the data from nasopharyngeal-oral pairs were retained. To balance the accuracy and biological interpretability, the full-length data were analyzed at the species-level, while the V4-region data were analyzed at the genus-level.

## Bacteria culture and identification

Fresh specimens were processed and cultured immediately after collection. Swabs were vortexed in PBS solution and prepared into 1:10, 1:100 and 1:1000 dilutions. Saliva samples were mixed and diluted until 1:10,000. Fifty microliters of each dilution were inoculated onto three different agars: Columbia blood agar (CBA, OXOID, Cat #: CM0331B), Kanamycin-vancomycin CBA and Phenylethyl CBA. The inoculated media were incubated under anaerobic conditions generated by the AnaeroPack system (MGC, Cat #: C-01) for 3 days. Twenty to fifty colonies per specimen, if possible, were randomly picked and subcultured on CBA for purification. Species were identified using Matrix Assisted Laser Desorption/ionization Time-Of-Flight mass spectrometry (MALDI-TOF-MS, Bruker) as recommended by the manufacturer. PCR amplification and 16S rRNA gene sequencing using primers 27 F /1492 R were performed to complement and validate the species identification results.

## AP-Colony PCR

AP-Colony PCR was performed in a 25 μL mixture containing 12.5 μL PCR master mix (Yeasen Biotechnology, Cat #: 10102ES03*), 11.5 μL water and 1 μL random primer D8635 (5'-GAG CGG CCA AAG GGA GCA GAC-3'). A small fraction (approximately 0.5 mm²) of the purified colony was picked directly as the DNA template. Thermal cycling was performed as follows: initial denaturation at 94 °C for 5 min; five cycles at 94 °C for 3 min, 37 °C for 3 min, and 72 °C for 3 min; then 30 cycles at 94 °C for 1 min, 55 °C for 1 min, and 72 °C for 3 min; and final extension at 72 °C for 10 min. Amplified products were electrophoresed in 1% GelRed-stained agarose gels and photographed under UV light. An 8-kb DNA ladder (Rubow, Cat #: QDRB-MK8) was used as the DNA size marker.

## Whole-genome sequencing and analysis

Isolates were grown in Brain Heart Infusion broth (BD, Cat #: 237500, Anaerobes Systems) at 37 °C for 2 days, and DNA was extracted with DNeasy PowerSoil kit. Libraries were prepared using ALFA-SEQ DNA Library Prep Kit (magigene, Cat #: M1010) and sequenced on Illumina NovaSeq 6000 platform. Paired-end reads were trimmed using Trimmomatic (v0.39)[45] and aligned using Bowtie2 (v7.3.0)[46]. For *Prevotella intermedia* and *Fusobacterium nucleatum*, *Prevotella intermedia* strain ATCC 25611 and *Fusobacterium nucleatum* strain ATCC 25586 were used as reference genomes, respectively. InStrain (v1.0.0) was utilized to assess the ANI between isolations[47]. The whole genomes were assembled using SPAdes (v3.15.5)[48] and annotated using PROKKA (v1.3)[49]. For phylogenetic analysis, representative genomes and annotations for the *Prevotella* and *Fusobacterium* genera were downloaded from ATCC database. Genus-level core genes were identified using Roary (v3.11.2) and phylogenetic trees were constructed[50].

## Meta-transcriptome sequencing and analysis

The meta-transcriptomic dataset used here has been described in detail in our previous study[38]. Briefly, 89 NPC tissues were collected from NPC patients during biopsy procedures and 12 non-tumor tissues were donated from patients with ear, nose and throat (ENT) benign diseases during their surgical procedures. These specimens were collected between 2018 and 2020 at SYSUCC. Total tissue RNA was extracted using TRIzol Reagent (Invitrogen, Cat #: 15596018). The sequencing libraries were generated with TruSeq® RNA LT Sample Prep Kit v2 (Illumina, Cat #: RS-122) and Ribo-Zero Gold kit (Illumina, Cat #: 20040526), and then sequenced using Illumina NovaSeq 6000 instrument. For microbiota detection, raw reads were processed with KneadData (v0.10.0, https://github.com/biobakery/kneaddata) to remove low-quality and human-derived sequences. Taxonomic identification sequences were classified using Kraken2 (v2.1.1)[51] against the constructed database, then the abundance of identified species was calculated by Bracken (v2.5.0)[52]. The species with an abundance

greater than one part per million were considered detectable. Meanwhile, reads that passed quality control were aligned to the human reference genome (UCSC hg38) using Hisat2 (v2.1.1)[53], and gene expression quantification was generated via htseq-count (v2.0.1)[54]. The differentially expressed genes (DEGs) were analyzed by edgeR (v3.36.0) with a significance cut-off of adjusted $P$ value less than 0.05 and $|\log_2FC|$ greater than 1[55]. Biological process (BP) of gene ontology (GO) and KEGG enrichment analysis were performed based on DEGs using ClusterProfilter (v4.2.2)[56]. Single sample Gene Set Enrichment Analysis (ssGSEA) was utilized to calculate the enrichment scores of immune cell types in the tissue microenvironment[57].

## Fluorescence in situ hybridization

Five-micron-thick paraffin-embedded sections were prepared and hybridized using standard techniques. The sequence of the universal bacterial probe (EUB338, Cy3 labeled) was 5'-GCT GCC TCC CGT AGG AGT-3'[15]. The probe sequences specific for two species were as follows: FITC-5'-ATG TTG TCC CTA VCT GTG AGG C-3'-FITC for *F. nucleatum*[58] and FITC-5'- CGT TGC GTG CAC TCA AGT C-3'-FITC for *P. intermedia*[59]. The slides were visualized and recorded using the OLYMPUS FV1000 microscope.

## Statistics and reproducibility

No statistical method was used to predetermine sample size, and we included sufficient samples with reference to other microbial studies. The samples without sufficient amounts of ASV sequences were excluded, as shown in Supplementary Fig. 1. Randomization and blindness were not performed in the study because statistical analyses relied on disease information. Student's $t$ test or Wilcoxon rank-sum test was used for continuous variables according to the data distribution. The association between continuous or rank variables was performed using Spearman's correlation test. The rates of categorical variables were compared by Fisher's exact test. $P$ value less than 0.05 was considered statistically significant. The false discovery rate (FDR) was used to adjust $P$ values in multiple hypothesis tests. Alpha diversity and data saturation were evaluated by rarefaction curve using USEARCH (v10.0.240)[60]. The differences in microbial communities were measured using the Bray-Curtis distance matrix and visualized by PCoA. Community differences between NPC and control were calculated using permutational multivariate analysis of variance (PERMANOVA)[61], the intergroup similarity between paired-sites for each participant was assessed using the Wilcoxon rank-sum test. Discriminatory bacteria were identified using ANCOM-BC (v1.0.5), correcting for sex, age, smoking, drinking, caries and other oral or nasal diseases[62]. Microbial interaction networks were analyzed using SparCC[63], and the microbes with significant correlations (correlation coefficient $r > 0.25$ or $< -0.25$, $P < 0.05$) were shown with Cytoscape (v3.9.0)[64]. Source tracking analysis of nasopharyngeal microbiota from oral cavity within the same participant was performed using FEAST (v0.1.0)[65] and SourceTracker2 (v2.0.1)[66]. The risk calculation of different levels of bacterial translocation was calculated using both the unadjusted and adjusted logistic regression models with adjusting the factors of age, sex, cigarette smoking status, alcohol drinking status, as well as the presence of caries and oral/nasal diseases. All statistical analyses were two-sided and executed using R version 4.1.2.

## Reporting summary

Further information on research design is available in the Nature Portfolio Reporting Summary linked to this article.

## Data availability

The metagenomics data generated in this study have been deposited in the National Center for Biotechnology Information (NCBI) under BioProject number PRJNA1011041. The whole genomics data of *Fusobacterium nucleatum* have been deposited in the NCBI under

BioProject number PRJNA1012570. And the whole genomics data of *Prevotella intermedia* have been deposited in the NCBI under BioProject number PRJNA1012572. The authenticity of this article has been validated by uploading the key raw data onto the Research Data Deposit platform (www.researchdata.org.cn), with the approval RDD number as RDDB2023382491. Source data are provided with this paper.

## Code availability

The code developed for data analysis is available at https://doi.org/10.5281/zenodo.10083512[67].

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

## Acknowledgements

We thank the staff and volunteers of Wuzhou Red Cross Hospital, Sun Yat-sen University Cancer Center and the First Affiliated Hospital of Sun Yat-sen University for their assistance in this study. We thank Wen-Hui Jia for the help in drawing model diagrams. This research was funded by the National Natural Science Foundation of China (grant number: 81973131 to W.H.J., 82273705 to W.H.J. and 82003520 to T.M.W.), the National Key Research and Development Program of China (grant number: 2021YFC2500400 to W.H.J.), the Basic and Applied Basic Research Foundation of Guangdong Province, China (grant number: 2021B1515420007 to W.H.J.), and Cancer Innovative Research Program of Sun Yat-sen University Cancer Center (grant number: PT19010701 to W.H.J.).

## Author contributions

W.H.J. conceived and designed the study, edited and reviewed the manuscript. Y.L. and Y.X.W. contributed to study design and manuscript writing. Y.L., M.Z.T., X.H.Z., M.Q.Z., Y.J.J., T.Z., and X.Z.L. recruited participants, collected samples and prepared baseline information. Y.L., Y.X.W., Y.D., Y.W.C., J.R.X., Q.Y.L. and X.T.T. completed sample processing and experiment. Y.L., Y.X.W., T.M.W., Y.Q.H., W.Q.X., D.W.Y. and H.D. conducted data analysis and interpretation. All the authors reviewed and approved the manuscript.

## Competing interests

The authors declare no competing interests.
