## [Peer Review File · Nature Communications]

Reviewers' Comments:

Reviewer #1:

Remarks to the Author:

The article presents an interesting investigation into the relationship between oral microorganism and nasopharyngeal carcinoma (NPC) through paired analysis (16SrRNA, culture and meta-transcriptomics) of nasopharyngeal and oral microbial samples in two cohorts (148 NPC patients and 124 non-malignant controls). Furthermore, the relationship between oral-translocated microbes and EBV loads, which is implicated in NPC, is explored, further demonstrating the potential role of *F. nucleatum* in cancer pathogenesis. A strength of the work is the use of nasopharyngeal biopsy to confirm the intratumoral presence of the oral microbes and functional analysis.

There are questions regarding the inclusion of the two cohorts. While it is great to have two unrelated groups, they do seem to have been treated differently, for example with different sequencing methods applied (full length 16S cohort 1 versus the much less specific V4 cohort 2). Could the authors comment on the impact of this and whether the data was combinable for species identification? Data seems to have only been presented for Cohort 1 (e.g. Figure 1). Also were culture-based work to validate species identification from a selection of the participants (n = 35) all from one cohort or both, and how were these individuals selected?

Meta-transcriptomics: The inclusion of meta-transcriptomics of the tumoral tissue provides direct and powerful information. At present it is unclear why the sample set is so unbalanced (89 NPC tumoral and 12 normal tissues) and also how these were selected/from which cohort. Also, given it would be expected to not have microbiota in the normal tissue biopsy, was findings from the control treated as background? And further to this, were sampling controls sequenced to check from background issues?

Given the above points on cohorts and methods, it would be useful for the authors to clarify which data and why has been included in the different analyses.

Below are specific comments:

- Figure 1e: To compare the phylogenetic trees of NPC and controls, this would be better displayed as a single tree and have prevalence overlaid as two bars per taxa in the outer ring circles?
- Intratumoral microbiota – I may have missed this, but was the entire listing of species and prevalence of microbes isolated from the NPC tumours provided?

Reviewer #2:

Remarks to the Author:

Please see the attachment.

Reviewer #3:

Remarks to the Author:

In the manuscript, the authors collected paired oral and nasopharyngeal samples in patients with nasopharyngeal carcinoma and performed microbiome association analyses. The major finding was that several oral taxa were enriched in the microbiome of nasopharyngeal cavity, which suggested potential bacterial translocation from the oral cavity. The translocated bacteria were further found inside of tumors and associated with Epstein-Barr virus infection. This is an interesting finding that adds to the exiting literature about microbiome-disease associations.

Here are my comments:

1. The authors found more similar microbiome compositions between oral and nasopharyngeal sites in NPC patients compared to controls. Their interpretation seems to be "translocation of oral pathobionts reshaped microbiota" (title, line 2). It is unclear to me when the patient samples were

collected. Did they receive any treatment before sample collection? If yes, it might be the treatment that caused the loss of niche specification and reshaped the microbiota. It will clear my concern if these patients were treatment naive at the time of sample collection.

2. Even though these patients were treatment naive, the observed relative enrichment of oral bacteria may not be driven by translocation. Higher relative abundance of oral bacteria may not indicate increased translocation, but may be caused by depletion of commensals. The differences between relative and absolute abundances matter for the interpretation. Are there significant differences in total bacterial load of nasopharyngeal microbiome between cancer patients and healthy controls?

3. Fig. 1b: NPC or control or both? It does not make sense to combine NPC and control for this plot.

4. Line 117-118: Were PCoA applied to NPC and control samples separately? If yes, they need to be separated so that their distances can be compared on the same set of axes.

5. The minimum report of statistics should include effect size, p value, sample size, and test method.

6. Fig. 1e and lines 121-123: The cladogram is unnecessarily complex and difficult to read. Why not show their differences in simple box plots?

7. Line 126: is bacterial translocation between oral and nasopharyngeal sites unidirectional or bidirectional?

8. Fig. 1f: there seems to be a big difference between SourceTracker2 and FEAST for translocation quantification. The authors divided all samples into high and low translocation groups based on FEAST. Why?

9. Line 133-137, 144-145, Table 1: How was the risk computed? What is the underlying statistical model? Details are needed in Method.

10. Line 212-219. What are the relative abundances of *Fusobacterium nucleatum* and *Prevotella intermedia* in the paired nasopharynx and saliva samples? In addition, AP-PCR is not quantitative. Why not sequence their whole genomes and compare the genome similarity? 99.999% ANI has been adopted to identify identical strains. See https://instrain.readthedocs.io/en/latest/important_concepts.html?highlight=identical#thresholds-for-determining-same-vs-different-strains for details.

11. Line 539-542: This is not acceptable. All sequencing data and necessary clinical metadata for reproduction of this study must be made publicly available before acceptance.

12. Computer codes are not provided.

POINT-TO-POINT RESPONSE TO THE REVIEWERS' COMMENTS

Reviewer #1:

*The article presents an interesting investigation into the relationship between oral microorganism and nasopharyngeal carcinoma (NPC) through paired analysis (16SrRNA, culture and meta-transcriptomics) of nasopharyngeal and oral microbial samples in two cohorts (148 NPC patients and 124 non-malignant controls). Furthermore, the relationship between oral-translocated microbes and EBV loads, which is implicated in NPC, is explored, further demonstrating the potential role of *F. nucleatum* in cancer pathogenesis. A strength of the work is the use of nasopharyngeal biopsy to confirm the intratumoral presence of the oral microbes and functional analysis.*

Reply:

We thank the reviewer for the positive comments and appreciate his/her careful evaluation and constructive suggestions for our work.

1. There are questions regarding the inclusion of the two cohorts.

1.1 While it is great to have two unrelated groups, they do seem to have been treated differently, for example with different sequencing methods applied (full length 16S cohort 1 versus the much less specific V4 cohort 2). Could the authors comment on the impact of this and whether the data was combinable for species identification? Data seems to have only been presented for Cohort 1 (e.g. Figure 1).

Reply:

Thanks for your valuable feedback. We acknowledge the reviewer's astute observation regarding the specificity of species identification using 16S rRNA gene V4 sequencing compared to full-length sequencing methods. Consequently, in cohort 2, we analyzed the V4 sequencing data at the genus level, a decision that posed challenges when attempting to merge the two cohorts at the species level. While the genus-based data did not impact the analysis of the microbial composition or the assessment of microbial translocation, it did constrain our ability to conduct the species-level validation in cohort 2. We have thoughtfully deliberated upon the implications of employing distinct

sequencing methods and their impact on data integration in the revised manuscript. As outlined in our DISCUSSION. [Page 16: Line 401-405]

“The utilization of both 16S rRNA gene full-length and V4 region sequencing methodologies presents a hurdle in combining data for species identification. In light of this, we advocate for the adoption of consistent sequencing methods across studies to facilitate seamless data integration and validation analyses across diverse cohorts.”

We apologize for any prior confusion. It's important to clarify that cohort 2 data were employed for the specific purpose of validating the observed microbial translocation phenomenon and its correlation with NPC. These validation results have been meticulously detailed in our Extended Data files. To summarize, the outcomes from cohort 2 were found to be in agreement with those from cohort 1 across various key aspects, including their results were presented in Extended Data files. Specifically, their findings were consistent with cohort 1 in microbiota diversity (Extended Data Fig. 2b & 2d), the similarity between oral and nasopharyngeal microbiota (Extended Data Fig. 4) and the association between microbial translocation and NPC (Extended Data Fig. 5). These consistent findings provide robust evidence that the phenomenon of bacterial translocation can indeed be validated in an independent population.

1.2 Also were culture-based work to validate species identification from a selection of the participants (n = 35) all from one cohort or both, and how were these individuals selected?

Reply:

The samples utilized for our culture-based work were not sourced from either cohort 1 or cohort 2. Instead, we acquired freshly collected samples from an independent cohort to ensure the preservation of a greater number of viable bacteria. For your reference, the relevant passage in our manuscript [Page 17: Line 435-439] now reads as follows: “In Addition, for our culturomics study, we prospectively collected paired nasopharyngeal swabs and saliva samples from a separate cohort comprising 48 individuals. These samples were obtained during nasopharyngoscopy procedures conducted at SYSUCC between May 2022 and June 2022, as well as between July 2023 and August 2023”

2. Meta-transcriptomics:

2.1 The inclusion of meta-transcriptomics of the tumoral tissue provides direct and powerful information. At present it is unclear why the sample set is so unbalanced (89 NPC tumoral and 12 normal tissues) and also how these were selected/from which cohort. Given the above points on cohorts and methods, it would be useful for the authors to clarify which data and why has been included in the different analyses.

Reply:

We apologize for any confusion arising from our earlier description of the study population. The meta-transcriptomic dataset was compiled from independent samples spanning the year 2018 and 2020. It's important to note that obtaining normal nasopharyngeal tissues is inherently challenging, as these “normal tissues” were obtained from patients with benign ENT diseases. Consequently, fewer individuals were willing to provide consent for the donation of their normal, fresh nasopharyngeal tissues. We have included a description of the meta-transcriptomic dataset acquisition in the METHODS section of the revised manuscript [Page 21: Line 544-551].

Furthermore, it is worth mentioning that the conventional treatment for NPC primarily involves radical radiotherapy rather than surgical intervention. Given the typically limited size of nasopharyngeal cancer tissue, samples collected from a single patient often do not suffice for multiple experiments of microbial sequencing, culturomics and tissue-based transcriptome sequencing. As a result, we recruited independent populations at different stages of our study to ensure an adequate sample pool.

2.2 Also, given it would be expected to not have microbiota in the normal tissue biopsy, was findings from the control treated as background? And further to this, were sampling controls sequenced to check from background issues? Given the above points on cohorts and methods, it would be useful for the authors to clarify which data and why has been included in the different analyses.

Reply:

Thanks for the comment. we made a deliberate choice not to exclude the microbiota

present in “normal tissues” as background. This decision was guided by the recognition that the nasopharynx is an integral component of the upper respiratory tract and maintains an anatomical connection with the external environment. Moreover, previous literature has reported the presence of the resident microbiota in normal nasopharyngeal tissues [1, 2]. So, instead of treating them as background, we compared whether the translocated detected between the tumor and normal tissues. Results showed that the identified translocation species had a higher detection rate in tumors than normal tissues, such as *Fusobacterium nucleatum* (29.2% and 8.3%, respectively) and *Prevotella intermedia* (10.1% and 0%, respectively).

As pointed out by the reviewer, the meta-transcriptome studies ideally incorporate sampling controls, which were lacking in our study. However, we thought that the translocated microbes detected in the nasopharyngeal tissues were more likely to be genuine because (1) tissue sampling was carried out using the same operating procedures and following the principles of aseptic operation, and nucleic acid processing and sequencing were also performed in the same batch; (2) The presence of translocated microbes identified in this study is not uniform across the same batch of samples but only be detected in some samples; (3) they are not the common agent/laboratory contamination reported, and (4) their existence intra tumor was also validated by Fluorescence in situ hybridization (FISH). Data from well-established experimental design including sampling controls are better for interpreting intratumoral bacteria. We have addressed the limitation of the lack of sampling controls in the section DISCUSSION [Page 16: Line 408-410].

Below are specific comments:

- Figure 1e: To compare the phylogenetic trees of NPC and controls, this would be better displayed as a single tree and have prevalence overlayed as two bars per taxa in the outer ring circles?

Reply:

This is a nice suggestion. We hope that the revised Fig. 1e, presented as follows, will enhance its comprehensibility.

Revised Fig. 1e | The phylogenetic tree represented ASVs of core species with > 5% presence in nasopharyngeal microbiota. The bar plots indicated the relative abundance of these species in nasopharynx which the filled colors indicate the types of microbes (top three commensal genera of nasopharynx and oral cavity, or potential pathogens). The heatmap indicated the prevalence of microbes in nasopharynx.

- Intratumoral microbiota – I may have missed this, but was the entire listing of species and prevalence of microbes isolated from the NPC tumours provided?

Reply:

Thanks for your comments. We have added the entire listing of the identified species along with their prevalence in the NPC tumors and normal tissues. This information can now be found in Revised Supplementary Table 6.

Reviewer #2:

The authors have characterized the nasopharyngeal and oral microbiota in cohorts of NPC patients and normal people. A panel of bioinformatic analyses had been done and some positive correlations with events in NPC pathogenesis were observed. This study is informative for understanding the potential roles of microbes in contributing the whole tumor microenvironment.

Reply:

We thank the reviewer for the insightful comments, and appreciate the reviewer's expertise and the time he/she has taken to review our work.

- 1. In the part of "Translocation-related microbes influenced the EBV infection in nasopharynx", (Line 278 – 280) the authors inferred that abnormal nasopharyngeal microbiota alternations were closely associated with regional EBV infection. However, what is meant by 'regional EBV infection'? As a matter of fact, all NPC tumor cells contain episomes of EBV, while normal nasopharyngeal epithelial are seldom detected with EBV infection. Therefore, in figure 5a, the group with undetectable EBV load is consist of normal people, while the groups with low to high EBV load are mainly consist of NPC patients. It is expected that the numbers of detected translocated species or NPC-enriched species will be correlated with higher EBV load (i.e. the groups of NPC patients).*

Reply:

You are correct. The previous expression was not accurate, we have now made the necessary modification in the manuscript to read "the EBV infection in the nasopharynx" as of [Page 11: Line 285].

We acknowledge the reviewer's concern regarding the observed correlations between translocated microbes and EBV load. In response, we conducted a more focused analysis, examining the association between EBV load and microbiota separately in the NPC patient group and control group. In the NPC group, our analysis continued to reveal a significant association between the number of detected "NPC_OtoNP" microbes and EBV load (Revised Extended Data Fig. 6a). Despite the constraints of limited sample size, we still observed positive correlations between EBV load and the

specific translocated species, such as *Peptostreptococcus stomatis*, *Capnocytophaga gingivalis*, as illustrated in the Revised Extended Data Fig. 6b.

In the control group, EBV was detected in only 12% of the individuals, we further explored the association between EBV and microbiota association by categorizing individuals into two group: EBV load as EBV-positive or EBV-negative. Our analysis revealed that EBV-positive controls exhibited a higher presence of NPC_{0toNP} microbes compared to EBV-negative ones (Revised Extended Data Fig. 6c). Additionally, the results also showed a trend toward a higher prevalence of translocated microbes in EBV-positive individuals when compared to EBV-negative individuals (Revised Extended Data Fig. 6d).

Revised Extended Data Fig. 6 | The association between oral translocated microbes and EBV infection in the nasopharynx in the NPC patients or controls group.

We have made the necessary adjustments to the results description to accurately reflect our findings [Page 11: Line 269-270]. Furthermore, we have the limitations of the cross-

sectional study design and the constraints posed by our sample sizes in the DISCUSSION [Page 15: Line 379-382], reading as follows:

“Constrained by the cross-sectional study design, our study can only provide preliminary evidence of the correlation between nasopharyngeal microbiota and EBV load. To substantiate these findings, further prospective studies with larger sample sizes are warranted.”

2. *Besides, what is meant by “a higher EBV load” detected by a nasopharyngeal swab? If the swab caught more NPC cells, the load may be higher. Or in another situation, if some of the NPC cells are undergoing lytic reactivation, the EBV copy number can rise from <50 copies to hundreds or thousands of copies per cell.*

Reply:

Thanks for your valuable comments. The term “a higher EBV load” signifies a higher level of EBV numbers in the nasopharynx exfoliated cells. Our approach to sampling nasopharyngeal swabs followed the standardized procedures that have been evaluated in our previous series of studies [3-5]. Endoscopy was used to evaluate the entire nasopharynx to locate the suspicious tumor sites, the swabs were collected by rotating over the mucosal epithelium at the suspected lesion site. In the case of NPC patients, a significant proportion of these cells were indeed tumor cells [3].

To address concerns about cell numbers, we have implemented careful measures in our analysis. Specifically, we quantified the nasopharyngeal EBV load using two metrics: "EBV copies per nanogram of DNA" (as described in the manuscript) and "EBV copies per cell (measurement of β -globin)." Importantly, our results revealed a strong correlation between the EBV load measured by these two methods, with a Pearson correlation coefficient of 0.88. Furthermore, when using the “EBV copies per cell” as the unit, we still observed positive correlations between the numbers of translocated microbes and EBV load in both NPC and control groups (Spearman’s $r = 0.29$, $P = 0.018$). This correlation underscores the reliability of our EBV load measurements and addresses concerns regarding the potential variations in cell numbers and their impact on our results.

We agree with the reviewer that cells undergoing lytic reactivation can contribute to an increase in the EBV copy number. It is indeed common for NPC tissues to contain cells in both the EBV-lytic and EBV-latent phases. In our previous study, we also detected the expression of both the latent and lytic gene transcripts in nasopharyngeal swabs from NPC patients [5]. In the context of our study, the EBV DNA detected by nasopharyngeal swabs may originate from both latent- and lytic-phases cells.

3. *In the conclusion, Line 384 – 386, “our study uncovers the oral-to-nasopharyngeal microbial translocation and its associated with an increased risk of NPC”, however, it’s the migration of microbes promote the NPC pathogenesis? Or the microenvironment in the NPC tumor, which consists of cancer cells harbouring EBV and with aberrant INF signalings, that enables certain oral microbes growing in the nasopharynx?*

We appreciate the reviewer's comment. In our opinion, the colonization of migrated microbes, accompanied by the dysbiosis of the local microbial communities, may interact with the tumor microenvironment and participate in the occurrence and development of tumors. The translocation of oral microbes to extra-oral sites has been gradually observed in several cancers. Studies have reported that translocated oral microbes are present in precancerous lesions and are involved in precancerous-carcinoma processes [6-8]. For example, *Fusobacterium nucleatum*, was significantly enriched in colorectal adenomas, and its colonization could directly promote cell proliferation through the activating various signaling pathways such as Wnt/ β -catenin, or shape the pro-tumor microenvironment by regulating the secretion of pro-inflammatory cytokines indirectly [9, 10]. In addition, the immunosuppressive microenvironment of the tumors might also facilitate the colonization of certain microbes [11]. We agree that further prospective cohort studies and functional experiments are necessary to address the roles of microbial translocation in the NPC pathogenesis. In light of this, we revised our conclusion to better reflect our findings: “In conclusion, our study uncovers the oral-to-nasopharyngeal microbial translocation and its potential roles in NPC development” [Page 16: Line 414-415]. We have also

supplemented one paragraph of discussion to elaborate on this comment in the revised manuscript [Page 15-16: Line 383-395].

4. In the section about “validation of oral-to-nasopharyngeal microbial translocation phenomenon in NPC patients” (Line 190 – 203), the NPC_{OtoNP} microbe *Fusobacterium nucleatum* was only found in one NPC patients (Line 207). It is hard to conclude there is a microbial translocation phenomenon in NPC patients.

Reply:

We expanded the sample size in our culturomics study with the aim of corroborating the translocation of *Fusobacterium nucleatum*. Specifically, we have included an additional 13 pairs of nasopharyngeal-oral samples from NPC patients from July 2023 to August 2023 at SYSUCC. These samples were subjected to culturing procedures in accordance with our established protocols. Among them, we isolated *Fusobacterium nucleatum* strains from both the nasopharyngeal and oral samples from one patient (S0038). Subsequent analysis using AP-PCR and bacterial whole-genome sequencing confirmed that these isolates were indeed the same strain, denoted as *Fn.n1*. (Shown in Revised Fig. 3c&3e). Additional research and comprehensive investigation will be essential to further substantiate this conclusion. These results now were added in the revised manuscript accordingly [Page 9: Line212-215].

Revised Fig. 3 | c. AP-PCR patterns of *Fusobacterium nucleatum* from S007 and S038. e. The comparison of ANI between different isolates of *Fusobacterium nucleatum*.

Reviewer #3:

In the manuscript, the authors collected paired oral and nasopharyngeal samples in patients with nasopharyngeal carcinoma and performed microbiome association analyses. The major finding was that several oral taxa were enriched in the microbiome of nasopharyngeal cavity, which suggested potential bacterial translocation from the oral cavity. The translocated bacteria were further found inside of tumors and associated with Epstein-Barr virus infection. This is an interesting finding that adds to the exiting literature about microbiome-disease associations.

Reply:

We thank the reviewer for the positive comments and appreciate his/her careful evaluation and constructive suggestions for our work.

Here are my comments:

- 1. The authors found more similar microbiome compositions between oral and nasopharyngeal sites in NPC patients compared to controls. Their interpretation seems to be "translocation of oral pathobionts reshaped microbiota" (title, line 2). It is unclear to me when the patient samples were collected. Did they receive any treatment before sample collection? If yes, it might be the treatment that caused the loss of niche specification and reshaped the microbiota. It will clear my concern if these patients were treatment naive at the time of sample collection.*

Reply:

We apologize for our unclear description of the collection time of the samples. All the NPC patients were treatment-naïve at the point of sample collection. We have added this information in the METHODS [Page 17: Line 425].

- 2. Even though these patients were treatment naive, the observed relative enrichment of oral bacteria may not be driven by translocation. Higher relative abundance of oral bacteria may not indicate increased translocation, but may be caused by depletion of commensals. The differences between relative and absolute abundances matter for the interpretation. Are there significant differences in total*

bacterial load of nasopharyngeal microbiome between cancer patients and healthy controls?

Reply:

we appreciate the reviewer's insightful comments, particularly regarding the importance of distinguishing between relative abundance and absolute abundance in the interpretation of our findings. In response to this valuable feedback, we measured the total bacterial load using the qPCR method. We observed the total bacterial load between NPC patients and controls is comparable (Response Fig. 1), suggesting no significant differences in the absolute bacteria abundance between NPC patients and controls.

Response Fig. 1 | The total bacteria load of nasopharyngeal microbiome between NPC patients and healthy controls. *P*-value was evaluated by *t*-test. The sequences of amplified primers were as follows: 16S-universal-F (5'- GCA GGC CTA ACA CAT GCA AGT C-3'); 16S-universal-R (5'-CTG CTG CCT CCC GTA GGA GT -3'). Escherichia coli DNA was used to construct the standard curve for absolute quantification of bacterial load in the samples.

3. *Fig. 1b: NPC or control or both? It does not make sense to combine NPC and control for this plot.*

Reply:

Thank you for pointing this out. In the original manuscript, the violin plot of Fig. 1b was presented in both NPC and control. We have revised this figure to display separate violin plots for NPC and control groups (Revised Fig. 1b).

4. *Line 117-118: Were PCoA applied to NPC and control samples separately? If yes, they need to be separated so that their distances can be compared on the same set of axes.*

Reply:

To clarify, PCoA in Fig. 1c was initially conducted on the combined NPC and control samples, we plotted the figures separately in NPC and control to provide a better presentation of the difference in community distribution between the two groups. As shown in Fig. 1c, each plot in both NPC and control groups is on the same set of axes. We have added the corresponding methodological description in the figure legend of Fig. 1 [Page 32: Line 820-822].

5. *The minimum report of statistics should include effect size, p value, sample size, and test method.*

We are grateful to the reviewer for highlighting this important detail. We have thoroughly reviewed the entire manuscript and supplemented this information accordingly:

- A. Information on sample size has been supplemented in Page 11: Line 267, Page 30: Line 812, Page 32: Line 834, Page 34: Line 860-861 and Page 38-39: Line 903-904.
- B. Information on test methods has been supplemented in Page 2: Line 31, Page 6: Line 134, Page 6: Line 148, Page 7: Line 152, Page 7: Line 163, Page 11: Line 272, Page 32: Line 819-820, and Page 34: Line 847.
- C. Information on effect size and *p*-value has been supplemented in Page 6: Line 148.
- D. Supplemented the EBV-*Gemella haemolysans* association results which was mistakenly omitted in the original manuscript (Revised Fig. 5c and Page 11: Line 277).

6. *Fig. 1e and lines 121-123: The cladogram is unnecessarily complex and difficult to read. Why not show their differences in simple box plots?*

Reply:

Thank you for pointing this out. We hope the Revised Fig. 1e, presented as followed, will make it easier to understand.

Revised Fig. 1e | The phylogenetic tree represented ASVs of core species with > 5% presence in nasopharyngeal microbiota. The bar plots indicated the relative abundance of these species in the nasopharynx and the filled colors indicate the types of microbes (top three commensal genera of nasopharynx and oral cavity, or potential pathogens). The heatmap indicated the prevalence of microbes in the nasopharynx.

7. *Line 126: is bacterial translocation between oral and nasopharyngeal sites unidirectional or bidirectional?*

Reply:

Thank you for your insightful comments. In our opinion, the identified translocated species associated with NPC, such as *Fusobacterium nucleatum* and *Prevotella intermedia*, their primary direction of translocation is from the oral cavity to the nasopharynx. Our reasoning is supported by the following evidence: (1) the translocated species we identified are considered common oral microbes which are included in the expanded Human oral microbiome database (*eHOMD*); (2) these species were seldomly observed in the nasopharynx of healthy individuals (shown in Fig. 2g); (3) both the prevalence and the abundance were higher in the oral cavity than the nasopharynx (shown in Fig. 2g). While we acknowledge the existence of

bidirectional microbial translocation between oral and nasopharyngeal sites, which also was observed by the source-tracking analysis (Extended Data Fig. 5a&5b). It is in line with expectations due to the anatomical connection between these two sites,

Extended Fig. 5 | The total bacteria load of nasopharyngeal microbiome between NPC patients and healthy controls. Results from FEAST(a) and SourceTracker2 (b) algorithm.

We have discussed the direction of bacterial translocation in the DISCUSSION.

[Page 12: Line 307-312] “Since the nasopharynx is anatomically connected to the oral cavity, which provides the opportunities for microbial communication between these two sites. Our study revealed that oral pathobionts abnormally trans-colonized in nasopharynx and reshaped the local microenvironment, which was associated with NPC. Future studies are needed to explore the phenomenon and significance of bidirectional microbial translocation between these two sites in nasopharyngeal and oral diseases.”

8. *Fig. 1f: there seems to be a big difference between SourceTracker2 and FEAST for translocation quantification. The authors divided all samples into high and low translocation groups based on FEAST. Why?*

Reply:

We apologize for the inappropriate description of FEAST results as the translocation score. In fact, the high and low translocation groups were divided through a k-means method that took into account the results from both FEAST and SourceTracker2 algorithms (Fig. 1f). While there is a difference in the results obtained by these two algorithms, Spearman’s r is equal to 0.70 ($P < 0.0001$), suggesting a substantial level

of agreement between the two methods. In the revised manuscript, we have removed Supplementary Table 2 which only presented the result of the FEAST algorithm for grouping. The description of the corresponding results was modified accordingly [Page 6: Line 127-132].

9. *Line 133-137, 144-145, Table 1: How was the risk computed? What is the underlying statistical model? Details are needed in Method.*

Reply:

The odds ratio was calculated using both the unadjusted and adjusted logistic regression model with adjusting the factors of age, sex, cigarette smoking status, alcohol drinking status, as well as the presence of caries and oral/nasal diseases. We have added the details in the METHODS [Page 23: Line 591-594].

10. *Line 212-219. What are the relative abundances of *Fusobacterium nucleatum* and *Prevotella intermedia* in the paired nasopharynx and saliva samples? In addition, AP-PCR is not quantitative. Why not sequence their whole genomes and compare the genome similarity? 99.999% ANI has been adopted to identify identical strains. See https://instrain.readthedocs.io/en/latest/important_concepts.html?highlight=identical#thresholds-for-determining-same-vs-different-strains for details.*

Reply:

Thanks for your comments. To clarify our methodology, in our culturomics study, we employed selective culture media supplemented with kanamycin and vancomycin to isolate potential anaerobic oral pathobionts, including *Fusobacterium nucleatum*, from nasopharyngeal swabs that contained substantial facultative anaerobic bacteria. However, the colony counts resulting from this method are not indicative of the total bacterial load in the swabs. Therefore, using colony counts to determine the relative abundance of *Fusobacterium nucleatum* and *Prevotella intermedia* in the samples would be inaccurate. To provide a more precise assessment of the relative abundance of these microorganisms, we relied on our 16S rRNA sequencing data, which yielded the following results: the detection rate of *Fusobacterium nucleatum* in nasopharynx

and oral cavity was 12.8% and 94.9%, with a relative abundance of 0.71% and 1.14%, respectively. The detection rates of *Prevotella intermedia* in the nasopharynx and oral cavity were 7.05% and 72.4%, with a relative abundance of 0.41% and 0.79%, respectively.

We greatly appreciate the reviewer's insightful suggestions regarding strain identification. Following your suggestions, we conducted whole-genome sequencing for paired nasopharyngeal and oral isolates of *Fusobacterium nucleatum* and *Prevotella intermedia*. Using the inStrain software, we performed the average nucleotide identity (ANI) analysis and observed that the ANIs between isolates from the same individual's nasopharynx and oral cavity reached 99.999%. To further support our strain identification, we constructed phylogenetic trees based on core genes by software Roary. These phylogenetic analyses revealed that the isolates from the same individual's nasopharynx and oral cavity exhibited the closest evolutionary branch. Conversely, isolates from different individuals displayed relatively lower ANI values, approximately 95% and 97% for *Fusobacterium nucleatum* and *Prevotella intermedia*, respectively. These findings validated our strain classification results based on AP-PCR. These additional results have been integrated into the RESULTS section of the manuscript to provide a more comprehensive account of our strain classification methodology and findings [Page 9: Line 217-223].

Revised Fig. 3e-3h | (e, f) The comparison of ANIs between different isolates of *Fusobacterium nucleatum* (e) and *Prevotella intermedia* (f). (g, h) Phylogenetic trees for genus *Fusobacterium* and *Prevotella*.

11. Line 539-542: This is not acceptable. All sequencing data and necessary clinical metadata for reproduction of this study must be made publicly available before acceptance.

Reply:

We have submitted the newly generated raw sequencing data and relevant clinical metadata related to this study to the NCBI database. These data are accessible under BioProject numbers PRJNA1011041, PRJNA1012570 and PRJNA1012572. The patient's information and related data sets supporting the conclusions of this research have been deposited in the Research Data Deposit public platform (www.researchdata.org.cn) under accession number RDDB2023382491. All the information will be publicly available upon acceptance of the manuscript.

12. Computer codes are not provided.

Reply:

The code developed for data analyses has been deposited and can be accessed at the following Github repository:

https://github.com/Doria-xi/oral_to_nasopharyngeal_microbial_translocation

References

- [1] W.H. Man, W.A. de Steenhuijsen Piters, D. Bogaert, The microbiota of the respiratory tract: gatekeeper to respiratory health, *Nature reviews. Microbiology*, 15 (2017) 259-270.
- [2] D.W. Cleary, S.C. Clarke, The nasopharyngeal microbiome, *Emerg Top Life Sci*, 1 (2017) 297-312.
- [3] P.F. Zhang, X.H. Zheng, X.Z. Li, T. Tian, S.D. Zhang, Y.Z. Hu, W.H. Jia, Nasopharyngeal brushing: a convenient and feasible sampling method for nucleic acid-based nasopharyngeal carcinoma research, *Cancer communications (London, England)*, 38 (2018) 8.
- [4] X.-H. Zheng, L.-X. Lu, X.-Z. Li, W.-H. Jia, Quantification of Epstein-Barr virus DNA load in nasopharyngeal brushing samples in the diagnosis of nasopharyngeal carcinoma in southern China, *Cancer science*, 106 (2015) 1196-1201.
- [5] X.H. Zheng, R.Z. Wang, X.Z. Li, T. Zhou, J.B. Zhang, P.F. Zhang, L.X. Lu, W.H. Jia, Detection of methylation status of Epstein-Barr virus DNA C promoter in the diagnosis of nasopharyngeal carcinoma, *Cancer science*, 111 (2020) 592-600.
- [6] N.T. Baxter, M.T.t. Ruffin, M.A. Rogers, P.D. Schloss, Microbiota-based model improves the sensitivity of fecal immunochemical test for detecting colonic lesions, *Genome medicine*, 8 (2016) 37.
- [7] C. Lopes, T.C. Almeida, P. Pimentel-Nunes, M. Dinis-Ribeiro, C. Pereira, Linking dysbiosis to precancerous stomach through inflammation: Deeper than and beyond imaging, *Frontiers in immunology*, 14 (2023) 1134785.
- [8] C.H. Lo, D.C. Wu, S.W. Jao, C.C. Wu, C.Y. Lin, C.H. Chuang, Y.B. Lin, C.H. Chen, Y.T. Chen, J.H. Chen, K.H. Hsiao, Y.J. Chen, Y.T. Chen, J.Y. Wang, L.H. Li, Enrichment of *Prevotella intermedia* in human colorectal cancer and its additive effects with *Fusobacterium nucleatum* on the malignant transformation of colorectal adenomas, *Journal of biomedical science*, 29 (2022) 88.
- [9] J.Q. Liang, T. Li, G. Nakatsu, Y.X. Chen, T.O. Yau, E. Chu, S. Wong, C.H. Szeto, S.C. Ng, F.K.L. Chan, J.Y. Fang, J.J.Y. Sung, J. Yu, A novel faecal *Lachnoclostridium* marker for the non-invasive diagnosis of colorectal adenoma and cancer, *Gut*, 69 (2020) 1248-1257.
- [10] C.A. Brennan, W.S. Garrett, *Fusobacterium nucleatum* - symbiont, opportunist and oncobacterium, *Nature reviews. Microbiology*, 17 (2019) 156-166.
- [11] C.E. Atreya, P.J. Turnbaugh, Probing the tumor micro(b)environment, *Science*, 368 (2020) 938-939.

Reviewers' Comments:

Reviewer #1:

Remarks to the Author:

The authors have addressed all reviewer comments adequately. The updated manuscript, figures and supplementary tables improve the paper. No further comments.

Reviewer #2:

Remarks to the Author:

Comments #1:

In the result section of "The validation of oral-to-nasopharyngeal microbial translocation phenomenon in NPC patients", the research team has collected paired nasopharyngeal swabs and saliva specimens from 34 NPC patients and 14 non-tumor controls. However, colonies were only successfully cultured from five nasopharyngeal samples, and all of them are collected from NPC patients.

Therefore, in this experimental setting, no analyses can be performed between the "NPC patients" vs "non-tumor controls".

If summarizing the results stated in line 204 – 205, the conclusion that can be made in this culturomics test is only "There are presence of *Fusobacterium nucleatum* in both the oral and naso sites in two patients, and presence of *Prevotella intermedia* in both the oral and naso sites in three patients".

The authors should clearly define what kind of validation was achieved in this session.

Comments #2:

From the authors' replies to my previous questions (Question 1 to 2), the authors agree that "Regional EBV infection" is not an appropriate term to describe the EBV infection status in a human body. It is because, at the region of NPC tumor, all NPC tumor cells contain episomes of EBV, while normal nasopharyngeal epithelium is seldom detected with EBV infection. (Perhaps some memory B cells and some EBV-susceptible cells are also EBV positive.) Besides, when EBV infects a normal epithelial cell, the EBV may enter into lytic cycle (which gives rise to hundreds to thousands of EBV copy from one single cell). This contributes to the positive detection of EBV by PCR in some samples of the control group.

Moreover, the authors mentioned in the response letter that "The swabs were collected by rotating over the mucosal epithelium at the suspected lesion site. In the case of NPC patients, a significant proportion of these cells were indeed tumor cells." This also implies that some portions of the collected cells are not NPC cells, and the so-called "EBV load" is mainly reflecting how many NPC cells are collected in that sample.

Therefore, the conclusions that can be drawn from Figure 5 or the Extend Figure 6 would be ambiguous.

While it is likely that nasopharyngeal microbiota will interact with EBV infection in the pathogenesis of NPC, the authors may need to further tune down the claim of "microbiota influence EBV infection" if no further functional evidence could be provided.

Reviewer #3:

Remarks to the Author:

The authors have done a great job in addressing my comments. Congratulations! I do not have

any remaining question.

POINT-TO-POINT RESPONSE TO THE REVIEWERS' COMMENTS

Reviewer #1:

The authors have addressed all reviewer comments adequately. The updated manuscript, figures and supplementary tables improve the paper. No further comments.

Reply:

We truly appreciate your insightful comments during the revision process. Once again, we extend our heartfelt gratitude for your time and expertise throughout the review process.

Reviewer #2 :

Comments #1:

In the result section of “The validation of oral-to-nasopharyngeal microbial translocation phenomenon in NPC patients”, the research team has collected paired nasopharyngeal swabs and saliva specimens from 34 NPC patients and 14 non-tumor controls. However, colonies were only successfully cultured from five nasopharyngeal samples, and all of them are collected from NPC patients.

Therefore, in this experimental setting, no analyses can be performed between the “NPC patients” vs “non-tumor controls”.

*If summarizing the results stated in line 204 – 205, the conclusion that can be made in this culturomics test is only “There are presence of *Fusobacterium nucleatum* in both the oral and naso sites in two patients, and presence of *Prevotella intermedia* in both the oral and naso sites in three patients”.*

The authors should clearly define what kind of validation was achieved in this session.

Reply:

Thanks for your suggestion. This result section describes the *Fusobacterium nucleatum* and *Prevotella intermedia* translocation in NPC patients, validated via culturomics methods, complementing their identified in 16S rRNA sequencing datasets. As suggested, we have revised the subtitle to “Culturomics validated the

translocation of *Fusobacterium nucleatum* and *Prevotella intermedia* in NPC patients”, aiming for a clearer expression.

Comments #2:

From the authors’ replies to my previous questions (Question 1 to 2), the authors agree that “Regional EBV infection” is not an appropriate term to describe the EBV infection status in a human body. It is because, at the region of NPC tumor, all NPC tumor cells contain episomes of EBV, while normal nasopharyngeal epithelium is seldom detected with EBV infection. (Perhaps some memory B cells and some EBV-susceptible cells are also EBV positive.) Besides, when EBV infects a normal epithelial cell, the EBV may enter into lytic cycle (which gives rise to hundreds to thousands of EBV copy from one single cell). This contributes to the positive detection of EBV by PCR in some samples of the control group.

Moreover, the authors mentioned in the response letter that “The swabs were collected by rotating over the mucosal epithelium at the suspected lesion site. In the case of NPC patients, a significant proportion of these cells were indeed tumor cells.” This also implies that some portions of the collected cells are not NPC cells, and the so-called “EBV load” is mainly reflecting how many NPC cells are collected in that sample.

Therefore, the conclusions that can be drawn from Figure 5 or the Extend Figure 6 would be ambiguous. While it is likely that nasopharyngeal microbiota will interact with EBV infection in the pathogenesis of NPC, the authors may need to further tune down the claim of “microbiota influence EBV infection” if no further functional evidence could be provided.

Reply:

We appreciate the reviewer’s suggestion. In the revised manuscript, we have tuned down the claim about the microbiota influencing EBV infection. The title has been altered to exclude “EBV infection” and revised to “Microbes translocation from oral cavity to nasopharyngeal carcinoma in patients”, with corresponding adjustments made in the abstract.

Reviewer #3:

The authors have done a great job in addressing my comments. Congratulations! I do not have any remaining question.

Reply:

We are truly appreciative for your constructive feedback that has been invaluable in improving the quality of our work. We express our sincerest gratitude again for your time and expertise throughout the review process.